# Structures of co-transcriptional RNA capping enzymes on paused transcription complex

Yan Li [1,3], Qianmin Wang[1,3], Yanhui Xu [1,2] & Ze Li [1,2] ✉

The 5′-end capping of nascent pre-mRNA represents the initial step in RNA processing, with evidence demonstrating that guanosine addition and 2′-O-ribose methylation occur in tandem with early steps of transcription by RNA polymerase II, especially at the pausing stage. Here, we determine the cryo-EM structures of the paused elongation complex in complex with RNGTT, as well as the paused elongation complex in complex with RNGTT and CMTR1. Our findings show the simultaneous presence of RNGTT and the NELF complex bound to RNA polymerase II. The NELF complex exhibits two conformations, one of which shows a notable rearrangement of NELF-A/D compared to that of the paused elongation complex. Moreover, CMTR1 aligns adjacent to RNGTT on the RNA polymerase II stalk. Our structures indicate that RNGTT and CMTR1 directly bind the paused elongation complex, illuminating the mechanism by which 5′-end capping of pre-mRNA during transcriptional pausing.

Transcription stands as a fundamental biological process through which genetic information is transcribed from DNA to RNA, subsequently translating into protein for specific physiological functions[1,2]. Precursor messenger RNA (pre-mRNA), miRNA, siRNA, and other noncoding RNAs are transcribed by RNA polymerase II (Pol II)[3]. Co- and post-transcriptional modifications, including 5′ capping, splicing, and 3′ polyadenylation, are vital for maturing pre-mRNA into mRNA and distinguishing it from other RNA molecules.

In eukaryotes, the unique 5′ cap structure, m7GpppNm, not only protects nascent pre-mRNA from degradation by ribonucleases, such as DXO[4] and XRN[5], but also facilitates mRNA nuclear export and translation[6,7]. In metazoans, the cap structure m7GpppN, also known as Cap0, is stepwise synthesized by RNA guanylyltransferase and 5′-phosphatase (RNGTT) and RNA guanine-7 methyltransferase (RNMT) bound with its activator RAM[8]. RNGTT possesses a 5′-triphosphatase domain (TPase) that removes the γ-phosphate of 5′-triphosphate, and an RNA guanylyltransferase domain (GTase) that adds the GMP to the 5′-diphosphate, forming the 5′-GpppN structure. Subsequently, RNMT-RAM methylates the N7 position of guanine to form the Cap0 structure[9–11]. Specific Cap-specific mRNA (nucleoside-2-

O-)-methyltransferase 1 (CMTR1) and 2 (CMTR2) methylate 2′-O of +1 and +2 ribonucleotides, respectively, producing the m7GpppNm (Cap1) and m7GpppNmNm (Cap2) structures[12–14].

Dephosphorylation, guanylylation, and methylation do not occur until the nascent pre-mRNA protrudes from the RNA exit tunnel of Pol II. RNase footprinting assay shows that Pol II protects pre-mRNAs of 15 nucleotide (nt)[15], which has been visualized by the structures of transcription elongation complexes (EC) as well[16–18], indicating that 5′-end of RNA shorter than 15 nt is inaccessible for the capping enzymes (CE). Capping occurs at the 5′-triphosphate of the first transcribed nucleotide when it stretches out from Pol II[15,19,20]. It is also reported that capping does not occur until the transcript is elongated to 20 nt in vitro and 30 nt in vivo[21,22]. Therefore, the length of RNA for capping in mammals remains ambiguous.

Previous studies have suggested that the Pol II C-terminal domain (CTD) is involved in recruiting and activating the capping enzymes[23,24]. The conserved Pol II CTD consists of a tandemly repeated consensus sequence $Y^1S^2P^3T^4S^5P^6S^7$, which is subject to extensive phosphorylation. The Ser-5 phosphorylated Pol II CTD, catalyzed by CDK7, contributes to recruiting and stimulating the guanylylation activity of

[1]Fudan University Shanghai Cancer Center, Institutes of Biomedical Sciences, State Key Laboratory of Genetic Engineering and Shanghai Key Laboratory of Medical Epigenetics, Shanghai Medical College of Fudan University, Shanghai 200032, China. [2]The International Co-laboratory of Medical Epigenetics and Metabolism, Ministry of Science and Technology, China, Department of Systems Biology for Medicine, School of Basic Medical Sciences, Shanghai Medical College of Fudan University, Shanghai 200032, China. [3]These authors contributed equally: Yan Li, Qianmin Wang. ✉e-mail: zeli@fudan.edu.cn

GTase via directly binding with RNGTT[25–27]. The phosphorylated Pol II modestly stimulates the GMP-GTase intermediate formation for guanylylation. Besides, the CTD-independent capping activity of RNGTT has also been detected in vitro[28].

In vitro experiments demonstrate that the DSIF (DRB sensitivity-inducing factor) component SPT5 binds to the full-length RNGTT, with both the TPase and GTase independently interacting with SPT5[26,29,30]. SPT5 stimulates the guanylylation activity of GTase by several folds, yet it does not influence the 5′-triphosphatase activity[24,31]. The interaction between SPT5 and TPase domain has an allosteric effect on guanylylation activity of GTase. Although phosphorylated CTD and SPT5 individually stimulate capping activity, their combined effects do not result in a synergistic increase in guanylylation activity[31]. DSIF not only serves as an indispensable transcription factor during early transcription stages, especially from initiation to elongation, but also recruits the negative elongation factor (NELF) to Pol II to assemble into paused elongation complex (PEC) at the promoter-proximal region and keeps the transcription machinery in pausing state[32]. Pausing is one of the main rate-limited steps in transcription, providing a window for various regulatory and modification processes, including capping modification. In metazoans, transcribing Pol II stalls around 20–60 nucleotides downstream of transcription start sites (TSS) in nascent pre-mRNA[33,34]. The progression of capping from uncapped transcripts in the early pausing state (20–32 nt) to capped ones in the late pausing state (32–60 nt) illustrates that 5′ end capping and Pol II pausing are closely intertwined, emphasizing their spatiotemporal coupling[19].

Cap0 and Cap1 modifications occur within the nucleus, yet the genomic distribution of the three enzymes varies. Besides RNGTT, CMTR1 is enriched at the 5′-end of genes with its peak proximal to the TSS, but RNMT-RAM distributes along the entire pre-mRNA, which suggest CMTR1 possibly is more involved in transcriptional events at early stage compared to RNMT-RAM[11,35]. The C-terminal WW domain of CMTR1 interacts with the Ser-5 phosphorylated CTD[36], indicating that CMTR1 is closely associated with the transcribing Pol II, and CMTR1 may simultaneously dock on the Pol II with RNGTT.

In *Vaccinia* virus, a single trifunctional protein (D1) possesses triphosphatase, guanylyltransferase, and methyltransferase activities, forming a heterodimer with its activator D12 to produce the Cap0 structure in complex with vRNAP[37]. In *S. cerevisiae*, the capping enzyme is composed of triphosphatase Cet1 and guanylyltransferase Ceg1, which form a hetero-trimer or hetero-tetramer that binds to transcribing Pol II[38]. Although the structures of co-transcriptional capping enzymes in poxvirus and yeast have been determined, the components of mammalian co-transcriptional capping complex are known to be different. For metazoan capping enzymes, crystal structures of isolated domains and cryo-EM structures of capping enzymes on transcribing Pol II-DSIF complex have been determined[14,23,39]. However, none of them couples pre-mRNA capping with Pol II transcription pausing successfully.

Here, we employed *S. scrofa* Pol II, human DSIF, NELF complex, RNGTT, and CMTR1, and determined structures of PEC-RNGTT (EMD-37352 and PDB ID: 8W8E) and PEC-RNGTT-CMTR1 (EMD-37353 and PDB ID: 8W8F) complexes at nominal resolution of 3.53 Å and 4.0 Å, respectively. In the PEC-RNGTT structure, we unveil that RNGTT is docked adjacent to the Pol II stalk with the OB-fold inserting into the root of stalk, and positions TPase at the RNA exit tunnel. Moreover, we discover a different conformation of NELF in these complexes, which implies that the binding of capping enzyme relieves the NELF-mediated Pol II transcription regression. The PEC-RNGTT-CMTR1 structure demonstrates how RNGTT and CMTR1 are arranged around the Pol II stalk, close to the RNA exit tunnel. These structural evidences of capping enzymes co-existing with pausing factors illuminate the intricate relationship between 5′ end capping modifications and the transcriptional pausing of Pol II, enhancing our comprehension of their synergistic functions in transcription regulation.

## Results

### Assembly of the PEC-RNGTT complex
Regarding the temporal and spatial correlation of pre-mRNA capping with the early stage of transcription, previous studies have shown that Ser-5 phosphorylated Pol II CTD[23,24] and DSIF[24,31] stimulate capping enzymatic activity, and NELF-mediated Pol II pausing is tightly linked with capping events[19]. We first examined the factors known to regulate the RNGTT interaction with Pol II. Using the glycerol density gradient ultracentrifugation assay, we observed that the purified RNGTT protein co-migrated with Pol II that was subjected to CTD phosphorylation by human TFIIH[40], whereas no co-migration was observed for the unphosphorylated Pol II (Supplementary Fig. 1a, c). Considering the role of DSIF in stimulating capping enzymatic activity[24,31], we tested the role of DSIF in RNGTT binding with the phosphorylated Pol II in the absence and presence of NELF complex, respectively. DSIF alone did enhance Pol II and RNGTT interaction and migrated with Pol II-RNGTT to higher molecular-weight fractions. The addition of the NELF complex had no effect on RNGTT binding to Pol II (Supplementary Fig. 1b, d).

We assembled the complex by sequential addition of the phosphorylated Pol II, DNA-RNA scaffold, non-template strand, DSIF, RNGTT, and NELF complex. To stabilize the catalytic domain of RNGTT, we used the GTP analog GMPPNP. According to the length of pre-mRNA accommodated in the RNA tunnel in high-resolution Pol II structures[16–18,41], we chose 17 nt 5′-triphosphate RNA as pre-mRNA, which was successfully assembled into the paused elongation complex (PEC) in the presence of RNGTT for cryo-EM data collection.

### Structure determination of the PEC-RNGTT complex
The cryo-EM structure of the PEC-RNGTT complex was determined at a nominal resolution of 3.5 Å (Supplementary Fig. 2b, d) (EMD-37352). The Pol II core was resolved at 3.0 Å, whereas the peripheral DSIF, NELF, and RNGTT were solved at a local resolution ranging from 5 to 7 Å (Fig. 1b, Supplementary Fig. 2d and Supplementary Table 1). The structural model of PEC (PDB ID: 6GML)[41] was rigidly fitted into the corresponding densities, and manually adjusted in Coot (Supplementary Fig. 3a). Cryo-EM 3D reconstructions showed that RNGTT locates adjacent to the Pol II stalk, whereas the NELF complex surrounds the foot and funnel domains of Pol II.

Focused refinement notably improved the resolution of RNGTT to 5.8 Å (Supplementary Fig. 2b). The moderate resolution of RNGTT is consistent with earlier structural studies showing that the capping enzymes are highly mobile on Pol II[38]. The structural model of the GTase (PDB ID: 3S24)[23] was docked into the locally refined cryo-EM map, in which the density of the OB-fold was visualized (Fig. 1a and Supplementary Fig. 3b). The continuous density at the two-domain boundary facilitated the placement of the TPase model into the map (Fig. 1a and Supplementary Fig. 3c). We placed the models of TPase of mammalian RNGTT (PDB ID: 1I9S), GTase of RNGTT (PDB ID: 3S24), DSIF and NELF-B-A-C/D models (PDB ID: 6GML) into corresponding densities and adjusted the model in Coot. The N-terminal NELF-E model from AlphaFold2 were rigidly fitted in cryo-EM maps (Supplementary Fig. 3e, f). For the 17 nt pre-mRNA, 15 nucleotides were observed and modeled in the local refined map and the first two nucleotides at 5′-end were rigidly fitted (Fig. 1c and Supplementary Fig. 3d).

### Structure of the PEC-RNGTT complex
The structure of PEC-RNGTT shows that Pol II exhibits a conformation similar to that observed in the previous PEC structure[41]. DSIF wraps around the Pol II body from the upstream DNA exit to the RNA exit tunnel. NELF complex hangs at the periphery of the Pol II body (Fig. 1b, c and Supplementary Fig. 3a), which will be discussed below. RNGTT is located adjacent to the stalk of Pol II (Figs. 1b, c, and 2a). The OB fold, known for its affinity with oligonucleotide or oligosaccharide,

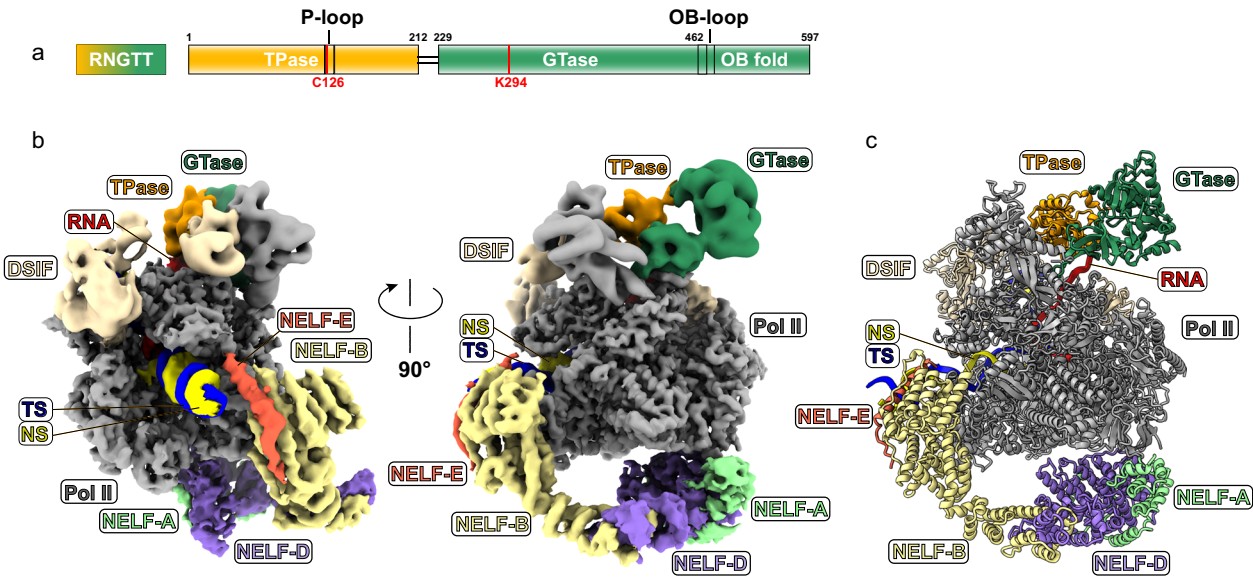

**Fig. 1 | Overall cryo-EM structure of the PEC-RNGTT. a** Domain architecture of RNGTT. The cryo-EM color scheme is used throughout all figures. Solid black lines represent the linkage region between TPase and GTase domains of RNGTT. The active site residues of the enzyme are marked in red lines. **b** Combined cryo-EM map of PEC-RNGTT in two different views and **c** Structural model of PEC-RNGTT in the same view. All components are depicted in the corresponding tab color.

mediates RNGTT-stalk interaction and stretches its OB loop (residue 476-484 aa) into the gap between RPB1 and the stalk (Fig. 2c). The OB-fold is stably positioned on the Pol II stalk and probably brings RNGTT in close proximity to the RNA exit tunnel, rather than the previously proposed Pol II foot domain[42].

In our PEC-RNGTT structure, the TPase domain not only tiles with GTase domain, but also interacts with the OB-fold through its N-terminus, forming a head-to-tail circular ring (Fig. 2b). TPase domain of RNGTT is surrounded by the OB fold, KOW2-3 of SPT5 and the Pol II stalk, with its catalytic pocket facing towards the RNA exit tunnel (Fig. 2d). While the N-terminal region of GTase detaches from the Pol II body, it still tethers to TPase and OB-fold (Supplementary Fig. 3c).

Previous structural studies show that KOWx-KOW4 and KOW5 form an RNA clamp at the RNA exit tunnel, suggesting a role in protecting nascent pre-mRNA for subsequent processing[16,41]. However, in our PEC-RNGTT structure, the association of RNGTT disrupts the original binding interface between KOWx-KOW4 and Pol II (Fig. 2e, f). The typical position of KOWx-KOW4 in PEC is occupied by the TPase domain and part of the OB fold, leading to diminished visibility of KOWx-KOW4 density. Despite these changes, the remaining portions of DSIF maintain positions similar to those observed in the PEC structure (Fig. 1b, c).

## PEC-RNGTT could efficiently guanylylate 17 nt RNA substrate in vitro

The precise length of RNA required for the initiation of capping reaction has been elusive for decades. In our cryo-EM structure, RNA protrudes from the RNA exit channel and reaches the TPase of RNGTT (Fig. 2a, d). Docking with the crystal structure of mouse RNA triphosphatase into TPase density[43], we observed that the 5′-triphosphate points toward the P loop of TPase. This loop contains a characteristic HCXXXXXRT motif and the catalytic residue Cys126, which are typical features of metazoan capping enzymes (Fig. 2d).

In the structure, 15 nucleotides were traced in the RNA exit channel, and the first two nucleotides at the 5′-end were not manually built, possibly due to the lack of stabilization (Fig. 2d and Supplementary Fig 4a). The DNA–RNA hybrid adopts a post-translocation conformation as that in EC, instead of the tilted conformation described in PEC (Fig. 2g and Supplementary Fig. 4b). Although the exposed

RNA density out of the RNA exit channel is not sufficient for unambiguous modeling, it points to the TPase domain, consistent with γ-phosphate dephosphorylation as verified by in vitro guanylylation assays (Fig. 2h). Based on the position of the RNA in our structure, we next performed guanylylation assay on RNAs in variable lengths (Supplementary Table 2). We observed that RNGTT efficiently catalyzed the guanylylation reaction on RNAs ranging from 17 to 20 nt. RNAs of 16 and 21–33 nt showed a decrease in guanylylation, and the reactions failed for 22 and 23 nt RNAs (Fig. 2h and Supplementary Fig. 4c). For 22 and 23 nt RNAs, the 5′-triphosphate may not be efficiently folded back to get access to the TPase active center and therefore lead to their failure of guanylylation by RNGTT. As RNA grows, the RNGTT-mediated guanylylation resumes. We speculate that the loop region linking the two domains of RNGTT and the sway of RNA may facilitate the enzymatic modules moving and catching the RNA substrates to complete enzymatic reactions.

## Non-canonical positioning of NELF in PEC-RNGTT

Cryo-EM 3D classification of PEC-RNGTT showed two different conformations of the NELF complex in the RNGTT-containing particles. One conformation mirrors previous observation in the PEC structure (canonical, termed "Up" state) (Supplementary Fig. 5). In contrast, the other presents a downward orientation towards the DNA duplex (noncanonical, "Down" state) (Fig. 3a). As we used NELF-D in our complex assembly, NELF-C/D module is referred as NELF-D in subsequent discussions.

Compared to the conformation in the "Up" state, the NELF complex conformation in the "Down" state shows noticeable differences. (I) The overall NELF complex shifts as far as 43 Å at the NELF-E N-terminal helix and rotates toward the entry DNA by approximately 35 degrees (Fig. 3c). (II) An undefined N-terminal region of NELF-E in the "Up" state displays a long helical density in the "Down" state map (Fig. 3c), which is well fitted with NELF-E (1-37 aa) Alphafold2 model (Supplementary Fig. 3f). It is a long positively charged helix possessing ten Lysine residues. The helix is positioned near the entry DNA. (III) Instead of dangling on the loop of the jaw domain in RPB5 in the PEC, NELF-B wraps the lateral surface of RBP5 and expands the binding interface (Fig. 3d). (IV) The NELF-A-NELF-D lobe rearranges intensively in the "Down" state compared with the "Up" state (Fig. 3e, f). In the

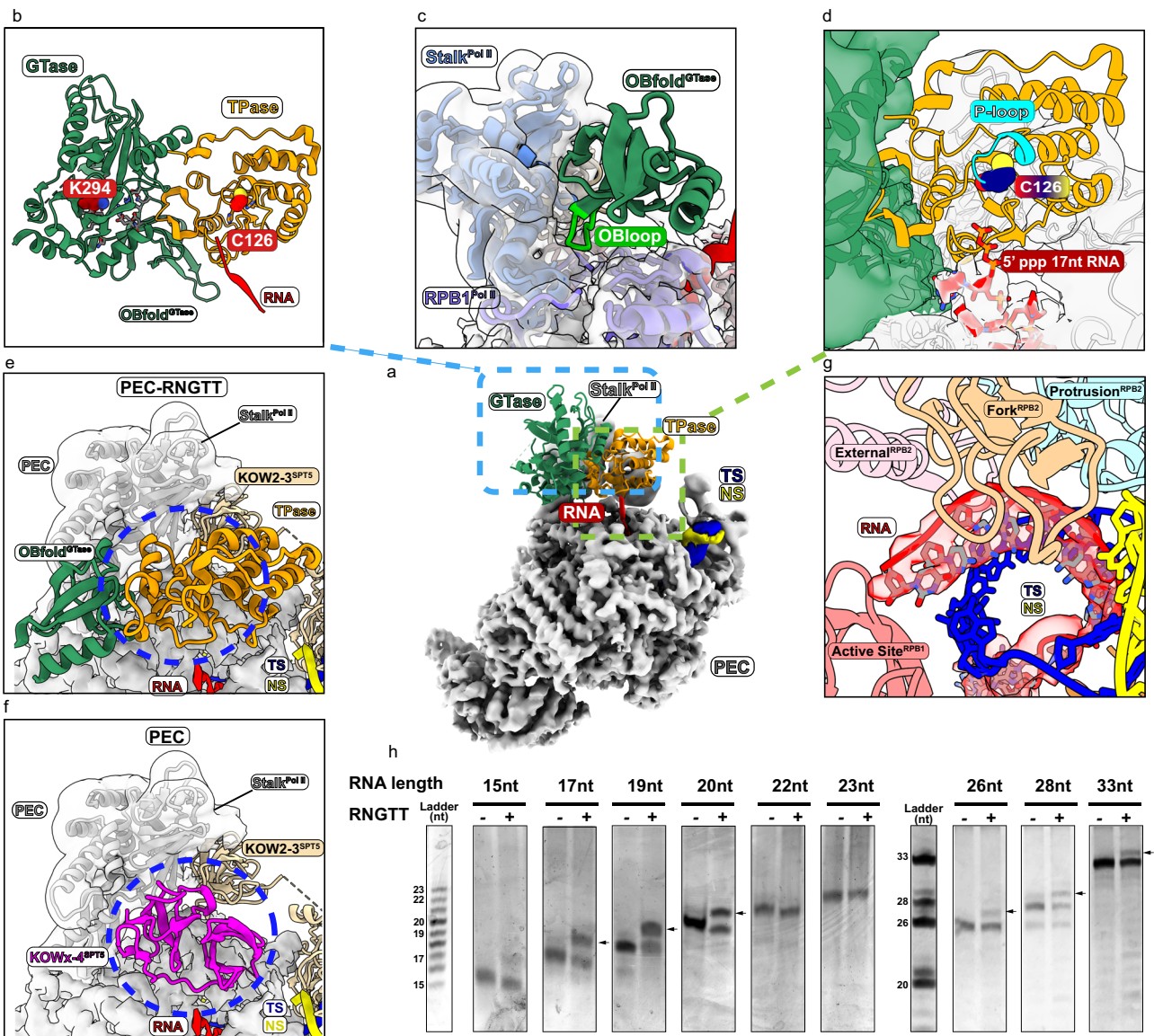

**Fig. 2 | RNGTT docks in close proximity to the Pol II stalk and catalyzes 5′-triphosphate RNAs in different lengths. a** Ribbon model of RNGTT docks on the PEC cryo-EM map. TPase and GTase domains of RNGTT are represented as ribbons in orange and sea green, respectively, while other components as density maps with labels in corresponding colors. **b** Close-up view of TPase and GTase domains of RNGTT. The active centers are displayed as spheres and colored by elements. **c** OB fold of GTase interacts with Pol II stalk (cornflower blue) and RPB1 subunit (medium slate blue). Pol II is illustrated as grey density map. OB loop of OB-fold is highlighted in lime. **d** TPase domain lies at the RNA exit tunnel. The active center of TPase is displayed as spheres and colored by elements. The triphosphate of RNA is displayed as sticks. Oxygen is colored in red and phosphorus is in orange. **e-f**

Comparison of PEC-RNGTT (**e**) and PEC (**f**) in the same view. (Pol II, grey; DSIF, wheat; KOWx-KOW4 domain, magenta). The blue ellipse circled the clash position between KOWx-KOW4 and RNGTT. **g** RNA in Pol II active center. Cryo-EM density of RNA is shown in semi-transparent red with its model fitted. Other components are showed as ribbons with labels in corresponding colors around. **h** In vitro guanylylation assay in the context of phosphorylated Pol II-DSIF-RNGTT complex. RNAs shorter than 23 nt correspond to the first ladder, while the others correspond to the second ladder. Bands of guanylylated RNAs are pointed out with arrows. The experiment was repeated at least three times. Source data are provided as a Source Data file.

canonical "Up" state, NELF-A is adjacent to RPB8, and NELF-D interacts with the Pol II funnel helices, cleft, and trigger loop, which collectively impede Pol II movement and retains Pol II pausing. Due to the rearrangement, the terminal portion of NELF-A-NELF-D lobe descends, and the middle helix bundle is lifted to interact with RPB8. The NELF-A tentacle, which is necessary for transcription pausing, vertically points to the middle part of the funnel helices (Fig. 3e). The C-terminal region of NELF-D detaches from RPB1 funnel and trigger loop, leaving the funnel partially open and convenient for NTP delivery to the active site, and trigger loop swings away from funnel helices, recovering its mobility under this conformation (Fig. 3f, g). In addition, the core module, containing RPB8 and RPB1 funnel, and the shelf module,

containing RPB1 cleft and foot domains, of PEC-RNGTT structure adopt a conformation similar to those of Pol II-DSIF and PEC structures, without relative movement between these two modules.

## Structural comparison with transcribing Pol II-DSIF-RNGTT complex

While preparing our manuscript, a related study by Garg et al. also investigated structures of co-transcriptional capping enzymes[39]. Garg et al. assembled RNGTT onto Pol II and the Pol II-DSIF-NELF complex using longer RNAs. In our study, we used identical components to construct the PEC-RNGTT complex. The main distinction lies in Garg et al.'s utilization of CDK7 to phosphorylate Pol II and their inclusion of

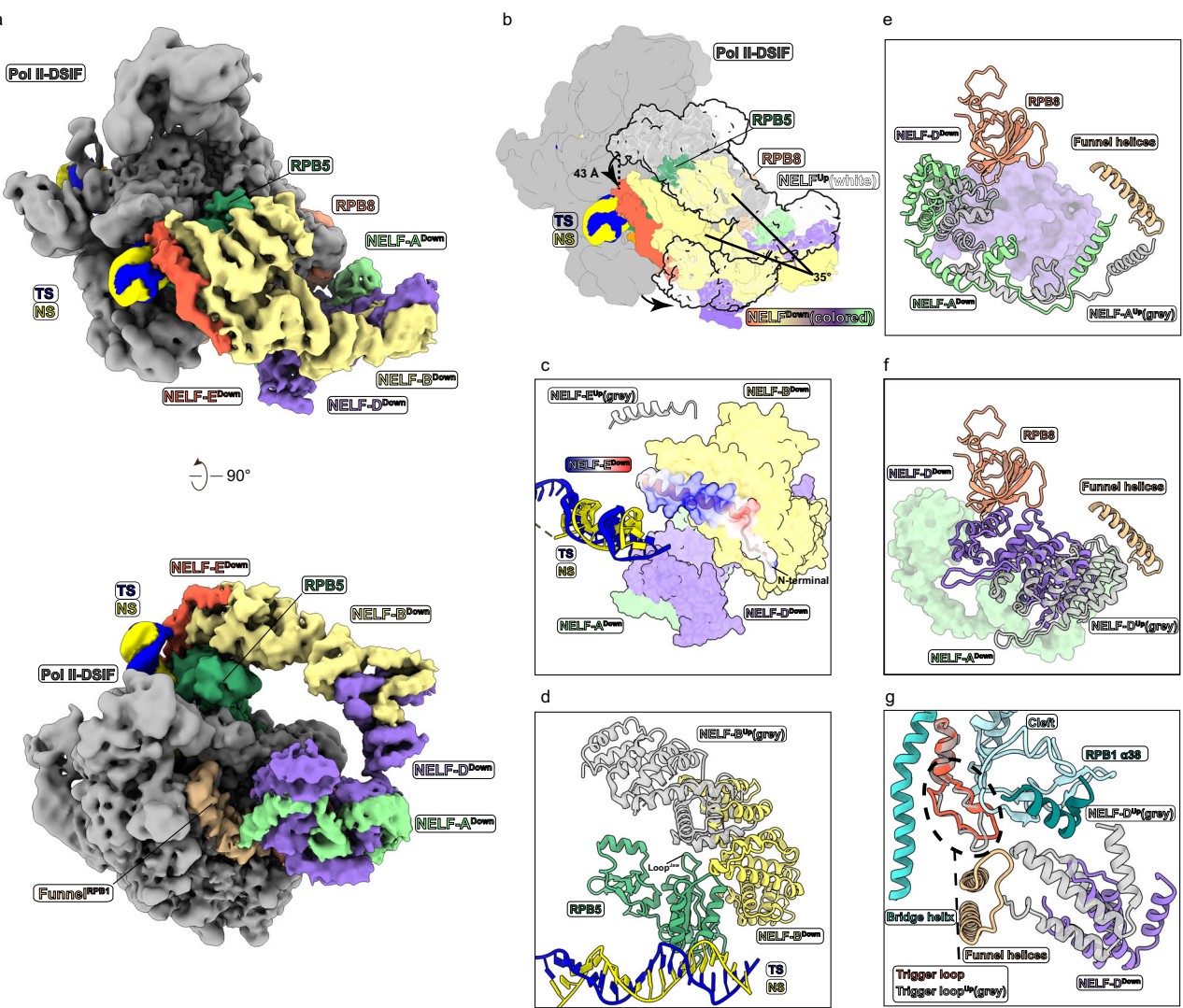

**Fig. 3 | Structure comparison of NELF in two conformations. a** Overall structure of NELF in "Down" state conformation of PEC-RNGTT in two views. The labels of different components are in corresponding map colors. **b** Comparison of the cryo-EM maps of NELF "Down" state (colored) and "Up" state (white and transparent). The conformational difference is indicated by moving distance and angle arrows. The distance is measured referring to NELF-E N-terminal helix and the angle is measured by NELF-B central axis. **c–g** Close-up views of NELF position comparison on the Pol II. NELF in "Down" state is colored, while NELF in "Up" state is grey and semi-transparent. **c** N-terminal helix of NELF-E^Down is in close proximity to DNA. N-terminal helix of NELF-E^Down is displayed as a charged surface. **d** Interface of NELF-B^Down and RPB5 of Pol II enlarges. **e** Interaction between NELF-A and NELF-D with RPB1 and RPB8 of Pol II changes a lot. NELF-A^Down is shown as ribbon, while NELF-D^Down is shown as a surface. **f** Another display form of NELF-A and NELF-D. NELF-D^Down is shown as ribbon, while NELF-A^Down is shown as surface. **g** NELF-D^Down moves away from the trigger loop and funnel helices of Pol II.

GTP as a substrate for guanylylation. Interestingly, Garg et al. did not observe particles containing both RNGTT and NELF. Instead, they detected two distinct conformations of RNGTT on the transcribing Pol II-DSIF complex. In one conformation (PDB ID: 8P4C), RNGTT is fully visible with RNA pointing towards the TPase domain, while in the other (PDB ID: 8P4D), TPase density is lacking, and RNA points towards the GTase domain (Supplementary Fig. 6b, c). Both studies identified the TPase located around the RNA exit tunnel, and the GTase anchored to the Pol II stalk via its OB loop (Supplementary Fig. 6a–c). However, Garg et al.'s structure highlighted the interaction of KOWx-KOW4 of SPT5 with the TPase domain, revealing conformational changes absent in our map (Supplementary Fig. 6d–f). Compared to our PEC-RNGTT structure (PDB ID: 8W8E), Garg et al.'s GTase domain in the TPase-invisible structure swings closer to the RNA exit tunnel (Supplementary Fig. 6a, c). We speculate that this swing is linked to the functional status of the GTase domain, as the GMPPNP used in our assembly might lock its active center, thus keeping the capping reaction

incomplete. This condition potentially allowed us to capture the NELF-containing capping complex. In summary, our PEC-RNGTT complex likely represents an earlier stage of the capping process compared to the structures previously reported by Garg et al.

## RNGTT and CMTR1 are arrayed on the PEC surface

CMTR1 has been reported to work at an early stage in pre-mRNA processing[35]. Glycerol density gradient ultracentrifugation analysis shows that CMTR1 in sub-stoichiometry co-migrated with phosphorylated Pol II, DSIF, and NELF in the absence or presence of RNGTT (Supplementary Fig. 7a, b). We thus prepared cryo-EM sample of PEC-RNGTT-CMTR1 and determined the structure at a nominal resolution of 4.0 Å with a periphery resolution ranging from 5.5 Å to 10 Å with no RNA density visible in CMTR1's catalytic pocket (Supplementary Fig. 8d) (EMD-37353 and PDB ID: 8W8F). The two conformations of NELF complex were observed in 3D classification (Supplementary Fig. 8b). We rigidly docked the crystal structure of RFM domain (PDB

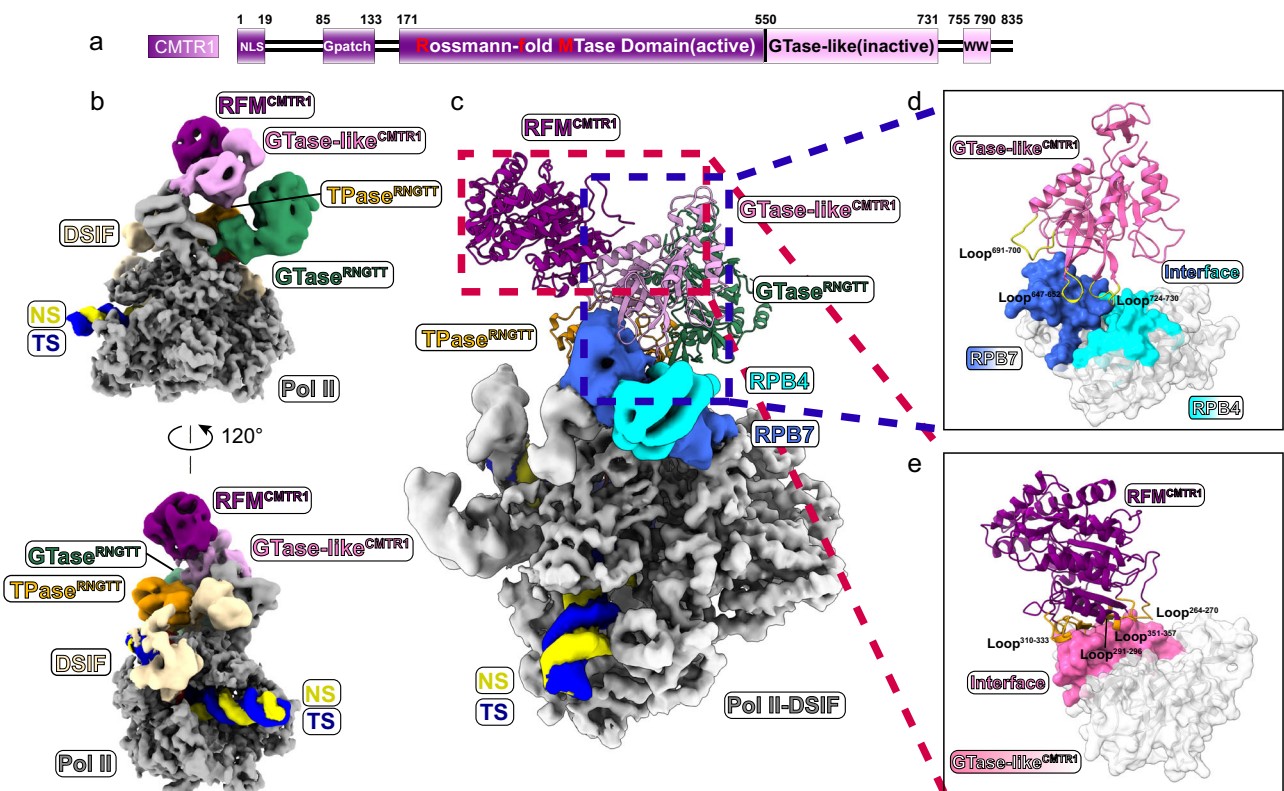

**Fig. 4 | Cryo-EM structure of the PEC-RNGTT-CMTR1 complex. a** Domain architecture of 2′-O-ribomethyltransferase, CMTR1. The cryo-EM map same color scheme is used throughout all figures. Solid black lines represent the linkage region between adjacent domains. **b** Combined cryo-EM map of PEC-RNGTT-CMTR1 complex. The labels of different components are in corresponding colors. **c** Ribbon models of RNGTT and CMTR1 dock on the PEC cryo-EM map. RFM and GTase-like domain of CMTR1 are displayed as models in dark magenta and plum, respectively, while other components are shown as density maps in corresponding tab colors. **d** Interaction between GTase-like domain of CMTR1 and Pol II stalk. Interface of RPB7 is highlighted as royal blue and interface of RPB4 is highlighted as cyan, while the other parts are white and transparent. Loops of GTase-like domain mentioned in the article are depicted as ribbons in yellow. **e** Interaction between RFM and GTase-like domain of CMTR1. Interface of GTase-like domain is highlighted as plum, while the other part is white and transparent. Loops of RFM domain mentioned in the article are shown as orange ribbons.

ID: 4N48)[14] and AlphaFold2 model of the GTase-like domain into CMTR1 cryo-EM map (Figs. 4a–c and Supplementary Fig. 3h). The structural models for other components of the complex were built as described above (Supplementary Fig. 3g).

In the PEC-RNGTT-CMTR1 structure, CMTR1 is anchored in the apical groove at the joint of RPB4 and PRB7 through flexible regions, mediated by apical loop regions of RPB4 and RPB7. Three loops (647-652 aa, 691-700 aa, 724-730 aa) from the GTase-like domain of CMTR1 contribute to CMTR1 docking on the Pol II tip (Fig. 4d), explaining why the deletion of GTase-like domain impairs its methylation activity[14]. The connection of RFM with GTase-like domain extends beyond the loop linkage between two domains. An intramolecular interaction generated by GTase-like domain and RFM domain anchoring loops (264-270 aa, 291-296 aa, 310-333 aa, 351-357 aa) further strengthens this connection (Fig. 4e).

The comparison of our results with the structures of Pol II-DSIF-RNGTT-CMTR1 (PDB ID: 8P4E) and Pol II-DSIF-CMTR1 (PDB ID: 8P4F)[39] suggests that CMTR1 remains bound to the stalk through the GTase-like domain while the RFM domain may rotate to capture and methylate the substrate. Our PEC-RNGTT-CMTR1 structure (PDB ID: 8W8F) captures the overall architecture of RNGTT, with the TPase situated adjacent to the stalk (Fig. 5a). This differs from the Pol II-DSIF-RNGTT-CMTR1 (PDB ID: 8P4E) and Pol II-DSIF-CMTR1 (PDB ID: 8P4F) structures[39], in which the TPase domain is invisible or RNGTT entirely absent (Fig. 5b, c). Superimposing the three structures reveal considerable differences, particularly in the orientation of CMTR1's RFM domain, which appears to rotate around the Pol II stalk by ~13 degree

relative to Pol II-DSIF-RNGTT-CMTR1 and ~30 degree relative to Pol II-DSIF-CMTR1 (Fig. 5d). The glycerol density gradient ultracentrifugation assays show that CMTR1 still co-migrate with PEC-RNGTT in absence of RNA or with 20-nt RNA, implying the binding of CMTR1 on Pol II is substrate-independent (Supplementary Fig. 7c, d). We thus speculate that the CMTR1 could pre-dock on Pol II in the absence of RNA substrate, and its flexibility allows CMTR1 to methylate the semi-capped pre-mRNA substrate. CMTR1 may occupy RNGTT's position, resulting in the latter's reduced visibility in the structure.

## Discussion

In this study, we determined the structures of PEC-RNGTT (PDB ID: 8W8E) and PEC-RNGTT-CMTR1 (PDB ID: 8W8F). We separated particles whose Pol II simultaneously bound with RNGTT and the NELF complex through 3D classifications. RNGTT was found adjacent to the stalk, while NELF was positioned around the foot and funnel domains of RPB1. However, no direct interaction between RNGTT and the NELF complex was observed in our structures.

While the interaction between RNGTT and the phosphorylated CTD was not captured in our structure, we observed that the OB-fold of RNGTT inserts into the gap at the base of the RPB4-RPB7 stalk. The GTase domain preceding the OB fold drifts outward. The configuration indicates that the mobility of GTase domain drives the transition between the open and closed conformations of RNGTT, rather than the OB fold, which is tightly clamped by the stalk and RPB1. This configuration provides insight into the observed change of OB-fold in the PBCV-1 capping enzyme[44]. The movement of GTase domain

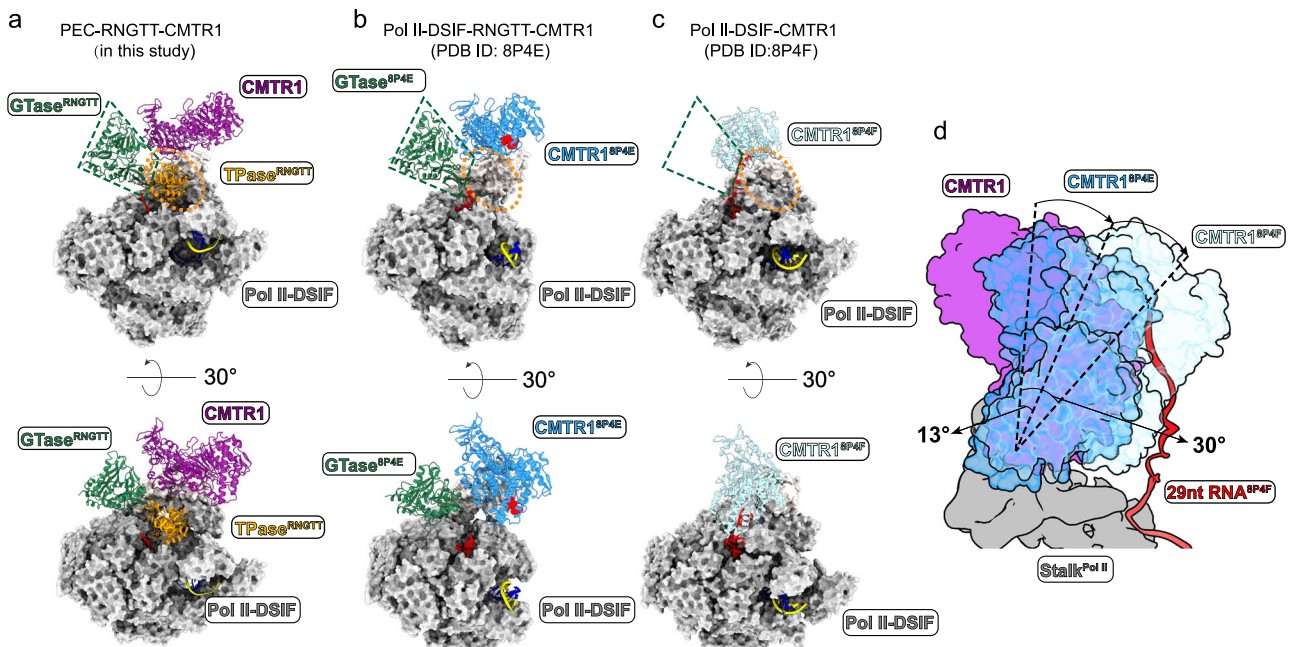

**Fig. 5 | Structure comparison of PEC-RNGTT-CMTR1, transcribing Pol II-DSIF-RNGTT-CMTR1 and transcribing Pol II-DSIF -CMTR1. a**–**c** Location of TPase[RNGTT], GTase[RNGTT], and CMTR1 of the three structures in two views. **a** PEC-RNGTT-CMTR1; **b** Transcribing Pol II-DSIF- RNGTT-CMTR1 complex (PDB ID: 8P4E); **c** Transcribing Pol II-DSIF- CMTR1 complex (PDB ID: 8P4F). **d** Orientation changes of CMTR1 in PEC-RNGTT-CMTR1 complex, 8P4E and 8P4F. CMTR1 in PEC-RNGTT-CMTR1 complex is colored in solid magenta, while CMTR1 in 8P4E and 8P4F in transparent sky blue and light blue, respectively.

facilitates the capture of the 5′-diphosphate RNA for guanylylation. We thus conclude that the interaction of OB-fold with Pol II is strong enough to stabilize RNGTT on Pol II, allowing it to adopt different conformations for its dephosphorylation and guanylylation activities. Moreover, the interaction between the N-terminal region of TPase and OB-fold facilitates the TPase catalytic pocket facing towards the RNA exit tunnel. The connecting loop region and structural interspace between the TPase and GTase domains do not impede the movement of GTase. Furthermore, compared with the PEC structures[41], TPase occupies the position typically held by KOWx-KOW4 in our structure, suggesting a displacement or conformational change of KOWx-KOW4. This placement enables TPase to catch the 5′-triphosphate of the nascent pre-mRNA.

The reported RNA length of the capping substrate varies in different capping assays. We set up a recombinant guanylylation assay in vitro, demonstrating that a 16-nt RNA is long enough to be processed in the presence of DSIF. However, this length is not exclusive, TPase is capable of catalyzing the dephosphorylation as long as its P loop can catch the substrate. We used in vitro transcribed RNA hybridized with DNA duplex as the scaffold to assemble the Pol II elongation complex in the presence of DSIF, without elongation or further purification of the complex. The catalytic ability of RNGTT is likely influenced by factors including nascent RNA accessibility, the mobility of two catalytic domains, and RNGTT binding affinity to the Pol II body.

In both PEC-RNGTT and PEC-RNGTT-CMTR1 cryo-EM maps, we observed two conformations of the NELF complex. Compared to the canonical conformation of the NELF complex in PEC[41], the other identified conformation is significantly different, particularly in the NELF-A-NELF-D lobe. The rearrangement of NELF-A and NELF-D helices dissociates NELF-D from the Pol II funnel and the trigger loop, facilitating a more open configuration that enhances NTP access. This conformational change implies the transitional state of Pol II from pausing to release or other subsequent events, such as cap binding[45]. We speculated that, upon RNGTT captures the 5′-end of pre-mRNA, DSIF or NELF-E would change its conformation and indirectly stimulates NELF-A-NELF-D reconstitution. Although biochemical data indicate that capping enzymes could promote pause release and transition from initiation to elongation[24,46,47], our structures did not establish a dynamic connection between RNGTT and the allosteric transition in NELF, leaving open the question how RNGTT influences on NELF conformation. Compared PEC-RNGTT with Pol II-DSIF EC and PEC structures, there is no conformational change of the core and shelf modules among them, which is also reported in PEC structure[16,41]. The swiveling of core module and shelf modules observed in reactivation from an arrested state in yeast Pol II, are not found in the reported PEC and our structures. We inferred that the disappearance of swiveling possibly results from the different factors and mechanisms involved in the pause-release and arrested-reactivation processes.

In our PEC-RNGTT-CMTR1 structure, CMTR1 firmly anchors at the tip of Pol II through the loop regions of the GTase-like domain, enabling the RFM domain to swing around and search for its substrate. Furthermore, the GTase-like domain also intra-molecularly interacts with the RFM domain. Considering its effect on CMTR1's enzymatic activity[14], the GTase-like domain likely stabilizes the RFM domain. During RNA elongation, the entire CMTR1 molecule wags around the anchoring point of the stalk and catches the substrate.

Comparing our study to the recently reported human co-transcriptional capping complex[39], we assembled both the PEC-RNGTT and PEC-RNGTT-CMTR1 complexes using the same components, including the transcription factors. For the PEC-RNGTT structure (PDB ID: 8W8E), we utilized a 17 nt RNA and observed the entire RNGTT anchoring in close proximity to the Pol II stalk. Furthermore, we detected two conformations of NELF in the PEC-RNGTT structures. For the PEC-RNGTT-CMTR1 structure (PDB ID: 8W8F), we visualized both the entire RNGTT and CMTR1 positioned adjacent to the Pol II stalk. The catalytic domain of CMTR1 is farther away from the RNA exit tunnel compared to the latest two structures of transcribing Pol II with CMTR1 (PDB ID: 8P4E and 8P4F)[39]. This discrepancy may be due to the longer RNA we used making it difficult for CMTR1 to catch and hinder us to find the RNA density. This suggests our model without RNA might

represent an initial position for CMTR1 docking on Pol II prior to substrate recognition.

Unfortunately, the methyltransferase RNMT-RAM failed to assembled into PEC or any other form of Pol II, leaving the relationships between RNMT-RAM and Pol II at different stages elusive. Given the RNA-dependent nature of RNMT-RAM functions, it may also require a suitable RNA at a specific Pol II stage. Due to sufficient space in our PEC-RNGTT-CMTR structure to accommodate extra proteins, the possibility remains that some unknown factors will help the RNMT-RAM association. Moreover, while NELF-E is known to play important roles in NELF recruitment, cap-binding of CBC, and 3′-end processing[45,48,49], the essential NELF-E tentacle, notably flexible, was only partially visible in our structures, limiting our ability to fully explore NELF-E's role in co-transcriptional capping process. In conclusion, our cryo-EM structures shed light on the co-transcriptional pre-mRNA capping and methylation processes, highlighting the association of capping enzymes with the Pol II paused elongation complex and revealing various conformational states of enzymes and transcription factors involved in transcription regulation.

## Methods

### Protein expression and purification
Full-length genes of human RNGTT and CMTR1 were cloned into pMlink vector containing a $4 \times$ ProteinA tag at the N-terminus and were transfected to Expi293F cells (Thermo Fisher Scientific, A14527) using PEI (Polysciences) when cell density is $2.5 \times 10^6$ /ml. Cells were harvested by centrifugation after cultured for another 72 h at 37 °C. Protein purification was performed at 4 °C. Cells were lysed for 30 mins in buffer A (30 mM HEPES-NaOH, pH 8.0, 300 mM NaCl, 10% (v/v) Glycerol, 0.25% (w/v) CHAPS, 5 mM $MgCl_2$, 5 mM ATP, 2 mM DTT (Dithiothreitol), 1 μg/mL Aprotinin, 1 μg/mL Pepstatin, 1 μg/mL Leupeptin. The cell lysate was clarified by centrifugation for 30 mins at $38,000 \times g$ using JLA-16.250 rotor (Beckman Coulter). The supernatant was kept and incubated with equilibrated IgG resin (Smart-Lifesciences) for 6 h and washed thoroughly with buffer B (30 mM HEPES-NaOH, pH 8.0, 300 mM KCl, 10% (v/v) Glycerol, 0.1% CHAPS, 2 mM $MgCl_2$, 5 mM ATP, 2 mM DTT). The target protein was released from the resin after on-column digestion overnight and eluted with buffer B. RNGTT was further loaded onto Mono Q (5/50GL, GE Healthcare), while CMTR1 was further loaded onto a Heparin column (5/50GL, GE Healthcare). Both proteins were eluted from the columns with NaCl concentration gradient. The peak fractions were collected and concentrated to -1 mg/mL, respectively. The concentrated samples were preserved at −80 °C for subsequent biochemical and structural analyses.

Transcription factors DSIF and NELF were purified in a similar way as describe in CEs. In brief, subunits (SPT4 and SPT5 for DSIF and NELF-A, -B, -D, -E for NELF) were subcloned into pCAG vector separately, and both SPT4 NELF-E were tagged with an N-terminal $2 \times$ Protein A. Plasmids for the same complex were co-transfected to Expi293F cells using PEI. The cells were collected and lysed in buffer A mentioned above. The supernatant was incubated with equilibrated IgG resin and washed thoroughly with buffer B. NELF was subjected to dephosphorylation with Lambda Protein Phosphatase (Beyotime, P2316S) before digestion in buffer containing 1 mM $MnCl_2$ at 4 °C overnight. The target protein was released from the resin after on-column digestion for 4 h and eluted with buffer B. Both DSIF and NELF were further loaded onto Mono Q and eluted from the columns with NaCl concentration gradient. The peak fractions were collected and concentrated to -2 mg/mL, respectively. The concentrated samples were preserved at −80 °C for subsequent biochemical and structural analyses.

Pol II was purified from *S. scrofa* thymus was prepared as previously described[41]. Briefly, 300 g thymus was homogenized in buffer A (50 mM Tris-HCl pH 7.4, 10% (v/v) Glycerol, 1 mM EDTA pH 8.0,

10 μM $ZnCl_2$) using a blender. The homogenate was clarified by centrifugation for 30 mins at $38,000 \times g$. The supernatant was kept and filtered with Miracloth (Millipore, 475855). PEI was added into the supernatant at a final concentration of 0.02%. The mixture was stirred for at least 10 min and subjected to centrifugation for 30 mins at $38,000 \times g$. The precipitate was kept and resuspended with buffer A containing 150 mM $(NH_4)_2SO_4$. The mixture clarified by centrifugation for 30 mins at $38,000 \times g$. The supernatant was kept and loaded onto Macro-Prep High Q Media (Bio-Rad). The media was washed thoroughly with buffer A containing 150 mM $(NH_4)_2SO_4$ and eluted with buffer A containing 400 mM $(NH_4)_2SO_4$. The elute was incubated with resin coupled by 8WG16 antibody, which is an antibody of Pol II CTD. The proteins were released by buffer A containing 500 mM $(NH_4)_2SO_4$ and 50% glycerol and loaded onto MonoQ column equilibrated with buffer C (30 mM HEPES-NaOH pH 8.0, 5% (v/v) Glycerol, 1 mM DTT, 2 mM $MgCl_2$, 0.2 mM EDTA, pH 8.0, 10 μM $ZnCl_2$). Pol II was eluted with NaCl concentration gradient. The peak fractions were collected and concentrated to -1.5 mg/mL. The concentrated samples were preserved at −80 °C for subsequent biochemical and structural analyses. There are four residue substitutions (G882S of RBP2, T75I of RPB3, S140N of RPB3, and S126T of RPB6) between *S. scrofa* and *H. sapiens* Pol II.

### Pol II phosphorylation
Ser5-phosphorylated Pol II was catalyzed by transcription factor TFIIH, which was prepared as described previously[50] and immobilized on IgG resin where the phosphorylation reacted. The TFIIH-contained resin was incubated with Pol II at 25 °C for 15 mins in buffer D (30 mM HEPES-NaOH, pH 8.0, 100 mM KCl, 5% (v/v) Glycerol, 6 mM $MgCl_2$, 50 μM ATP, 2 mM DTT for final concentration. Phosphorylated Pol II was eluted with buffer D and quantitated using SDS-PAGE gel stained by Coomassie blue. The protein was stored at −80 °C for further experiments. All the Pol II mentioned below was phosphorylated unless specially noted.

### Preparation of 5′-triphosphate oligoribonucleotides
5′-triphosphate oligoribonucleotides was synthesized by in vitro transcription using T7 High Yield RNA Synthesis Kit (YEASEN). Transcription template and HDV ribozyme with T7 promoter sequences were cloned into pUC57 vector and the extracted plasmids were further purified with Source Q 5/50 GL column (GE Healthcare). The DNA fractions were precipitated with 0.3 M NaAc and 70% (v/v) isopropanol and recovered (Sangon Biotech). The plasmids were linearized using XhoI (NEB) and extracted by phenol/chloroform (1:1). The final concentrations of the templates were measured about 1 μg/μl. All the steps below were RNase-free. According to the specification of the kit, in vitro transcription was performed. The transcription reaction was incubated at 60 °C for 30 mins to release target oligoribonucleotides from HDV ribozyme. The products were loaded onto 20% denaturing gel (7 M urea, $1 \times$ TBE, 20% Bis-tris acrylamide 19:1 gel). The target RNA bands were excised from gels and soaked in 0.3 M sodium acetate overnight at 4 °C. The RNA was precipitated with isopropanol. The precipitates were dissolved in DPEC water (Sangon Biotech) and analyzed with 15% native PAGE ($1 \times$ TG, 15% Bis-tris acrylamide 19:1 gel) gel and nanodrop.

### Sample preparation for cryo-EM
To obtain the DNA − RNA hybrid, template strand DNA (Generay Biotechnology) and 5′-triphosphate RNA were mixed with a molar ratio of 1:1.3 and were annealed following 95 °C for 5 mins and then decreasing the temperature by 1 °C min$^{-1}$ steps to 4 °C in a thermocycler in 20 mM HEPES-KOH, pH 7.4, 60 mM KCl, 3 mM $MgCl_2$, and 5% (v/v) glycerol. The nucleotide sequences used in complex assembly are as follows: template DNA: 5′-GCT CCC AGC TCC CTG CTG GCT CCG AGT GGG TTC TGC CGC TCT CAA TGG-3′, non-template DNA: 5′-CCA TTG AGA

GCG GCC CTT GTG TTC AGG AGC CAG CAG GGA GCT GGG AGC-3′, 17 nt RNA for PEC-RNGTT: 5′-GAG AGA GGG AAC CCA CU-3′, 40 nt RNA for PEC- RNGTT-CMTR1: 5′-AAU UAA GUC GUG CGU CUA AUA ACC GGA GAG GGA ACC CAC U-3′. All concentrations below referred to the final concentrations used in the complex assembling. Phosphorylated Pol II (120 pmol) was incubated with DNA-RNA hybrid at a 1:1.2 molar ratio for 10 mins at 30 °C. Hybrid containing 17 nt 5′-triphosphate RNA was used for PEC-RNGTT samples, and hybrid containing 40 nt 5′-triphosphate RNA was used for PEC- RNGTT-CMTR1 sample. Then non-template DNA was added at a 1:1.2 molar ratio with hybrid followed by another 10 mins incubation at 30 °C, producing an elongation complex (EC). Further assembly was stepwise and carried out at 25 °C for 20 mins. Four-fold molar excess of DSIF (480 pmol) was added first, and four-fold molar excess of capping enzymes (480 pmol), RNGTT alone or with CMTR1 were complemented subsequently. GMPPNP was added to a final concentration of 100 µM. Then four-fold molar excess of NELF (480 pmol) was added.

The assembled complexes were purified and crosslinked using gradient fixation (GraFix)[51]. The homogeneity of peak fractions was assessed by negative-staining electron microscopy. Qualified fractions were pooled, concentrated and replaced buffer to reduce the glycerol concentration under 0.5% (v/v). Negative-staining EM grids were prepared as previously described[52]. For cryo-EM grids preparation, 3 µL of the concentrated samples were applied to Quantifoil R 1.2/1.3 holey, 200 mesh carbon grids, which are freshly glow-discharged in the $H_2/O_2$ mixture for 30 s using a Gatan 950 Solarus plasma cleaning system with a power of 5 W. After incubation of 5 s at a temperature of 4 °C and a humidity of 100 %, the grids were blotted for 2 s with blot force −2 in a Thermo Fisher Scientific Vitrobot Mark IV and plunge-frozen in liquid ethane at liquid nitrogen temperature. The ø55/20 mm blotting paper (TED PELLA) is used for blotting.

## In vitro capping assay

In vitro capping assay was performed using purified RNGTT and EC containing 5′- triphosphate RNA as substrate. EC with different length RNAs (17 nt, 19 nt, 20 nt, 22 nt) was assembled as above, and the molar ratio between phosphorylated Pol II and hybrid was adjusted to 1.2:1. For one reaction, EC (3 pmol defined by hybrid) was mixed with RNGTT (8 pmol) in buffer containing HEPES-NaOH, pH 8.0, 100 mM NaCl, 5 mM $MgCl_2$, 2 mM DTT. The reactions were started with the addition of 100 µM rGTP at 37 °C for 60 mins and stopped by adding the same volume of stop buffer containing 7 M urea, 50 mM EDTA, and 1 × TBE. 1 µl of Protease K (Promega, 20 mg/ml) was added and digested for 30 mins at 30 °C. The reaction samples were analyzed by 20% denaturing gel (8 M urea, 1 × TBE, 20% Bis-Tris acrylamide 19:1 gel) in 1 × TBE buffer for 50 mins at 180 V constant. The gels were stained with SYBR™ Gold dye (Invitrogen, ThermoFisher Scientific) and visualized using Typhoon 9500 FLA Imager (GE Healthcare Life Sciences).

## Cryo-EM data collection

The cryo-EM data collection was finished with a Thermo Fisher Scientific Titan Krios transmission electron microscope operated at 300 kV. Cryo-EM images were automatically recorded by a Gatan K3 Summit direct electron detector equipped with a GIF quantum energy filter (Gatan) set to a slit width of 20 eV. All images were collected in the super-resolution counting mode using Serial-EM with a nominal magnification of ×64,000 in the EFTEM mode, yielding a super-resolution pixel size of 0.667 Å on the image plane. The defocus values ranging from −1.5 to −2.5 µm. Each micrograph stack was dose-fractionated to 40 frames with a total electron dose of ~50 e − /Å² and a total exposure time of 3.6 s. 15,281 micrographs of phosphorylated PEC-RNGTT and 10,690 micrographs of phosphorylated PEC-RNGTT-CMTR1 were collected for further processing.

## Image processing

Movie stacks were aligned by MotionCor2[53] with 5 × 5 patches and 2 × binned to a calibrated pixel size of 1.334 Å/pixel, generating drift-corrected micrographs with and without electron-dose weight. Contrast transfer function (CTF) parameters were estimated by Gctf[54] from non-dose-weighted micrographs. The particles were automatically picked by Gautomatch. The data processing were performed with RELION3.1[55,56] and cryoSPARC[57] v4 using dose-weighted images.

For the PEC-RNGTT dataset, the autopicked particles were extracted with 320³ pixels box size and rescaled the particles to 80³ pixels box size. The whole particle dataset was split into several sub-datasets for further reference-free 2D classification. The yielded particles were further subjected to the 3D classifications. The particles with good quality were rescaled to 160³ pixel box size and subsequently subjected to the 3D classification. This process provided two classes with different NELF conformations. The particles from the two NELF conformations were subjected for further 3D classification with different NELF module masks. A final set of 51,449 and 68,862 particles for NELF in "Up" state and NELF in "Down" state were selected to perform a final 3D reconstruction in cryoSPARC, yielding reconstruction of overall maps with "Up" state NELF and "Down" state NELF. Local refinement focused on NELF module with mask could reconstituted the NELF part at 4.62 Å and 4.83 Å, respectively. To improve the map quality of the RNGTT, the signal of RNGTT and partial of the Pol II stalk module were subtracted from six classes of 3D overall classification. The subtracted particles were subjected for further 3D classification by applying local mask for RNGTT and stalk module. The yielded particles were further subjected for particles subtraction with RNGTT mask. A 3D classification by applying a mask of the RNGTT resulted in a clean dataset containing 175,576 particles. The resulting particles were refined in cryoSPARC, yielding a reconstruction of RNGTT at 5.82 Å.

PEC-RNGTT-CMTR1 dataset were processed as described above.

## Model building and structure refinement

Model building was carried out by fitting the available cryo-EM and crystal structures of Pol II, NELF-B-A-C/D and DSIF from PDB: 6GML, the N-terminal of NELF-E from AlphaFold2 model, TPase of RNGTT from PDB: 1I9S, GTase PDB: 3S24 and RFM domain of CMTR1 from PDB: 4N48 into the EM density maps using UCSF Chimera. The models of GTase-like domain of CMTR1 and the long helix of NELF-E were predicted by AlphaFold2[58]. All the models were then manually adjusted in Coot[59]. The final model refinement was carried out using phenix.real_space_refine with PHENIX[60] and validated through examination of Ramachandran plot statistics, a MolProbity score. Model representations in the figures were prepared by PyMOL (http://pymol.org/) and UCSF ChimeraX[61].

## Reporting summary

Further information on research design is available in the Nature Portfolio Reporting Summary linked to this article.

# Data availability

The cryo-EM maps have been deposited in the EM Databank under accession numbers: EMD-37352 (PEC-RNGTT) and EMD-37353 (PEC-RNGTT-CMTR1). Atomic coordinates have been deposited in the Protein Data Bank with PDB IDs: 8W8E (PEC-RNGTT) and 8W8F (PEC-RNGTT-CMTR1). Previously published structures used in model building and structural comparison are available in the Protein Data Bank with PDB IDs: 6GML (Pol II-DSIF-NELF), 1I9S (the RNA triphosphatase domain of mouse mRNA capping enzyme), 3S24 (the mRNA guanylyltransferase domain of RNGTT), 4N48 (the RFM domain of CMTR1), 8P4C (Pol II-DSIF RNGTT), 8P4D (Pol II-DSIF RNGTT without the RNA triphosphatase domain of RNGTT), 8P4E (Pol II-DSIF-RNGTT-CMTR1) and 8P4F (Pol II-DSIF-CMTR1). Source data are provided with this paper.

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

## Acknowledgements

We thank Dr. Xizi Chen, Jiabei Li and Xinxin Wang for preparing Pol II from *S. scrofa* thymus. We thank Dr. Hai Zheng for RNA-DNA hybrid preparation. We thank the Center of Cryo-Electron Microscopy at Fudan University for the supports on cryo-EM data collection. This work was supported by grants from the National Natural Science Foundation of China (31630002, 32371251 to Z.L.) and Shanghai Pujiang Program (21PJD004 to Z.L.).

## Author contributions

Y.L. and Z.L. prepared Pol II in the apo form for structural analyses; Y.L. purified the other proteins with the help from Z.L.; Z.L. designed and prepared nucleic acid scaffold; Y.L. assembled the complexes, prepared cryo-EM samples and collected the data; Z.L. and Q.W. performed the cryo-EM analyses and the model buildings; Y.L. and Z.L. performed the capping assays and analyzed the data; Y.X., Z.L. and Y.L. designed the experiments and analyzed the data; Z.L.,Y.L. and Q.W. wrote the manuscript; and Z.L. supervised the project.

## Competing interests

The authors declare no competing interests.
