## [Peer Review File · Nature Communications]

REVIEWER COMMENTS

Reviewer #1 (Remarks to the Author):

In this manuscript, the authors report the cryo-EM structures of the mammalian paused Pol II elongation complex bound with the elongation factors DSIF and NELF, the capping enzyme RNGTT, which catalyzes the removal of the γ -phosphate of 5'-triphosphate and the addition of GMP to the 5'-diphosphate, and the capping enzyme CMTR1, which catalyzes the 2'-OH methylation of the +1 ribonucleotide. The cryo-EM structures present clear snapshots and structural mechanisms of the critical capping event during the initial stage of mRNA synthesis. Although the recent publication by Garg et al. (PMID: 37369200) also reported similar structures of mammalian co-transcriptional capping, the structures in this manuscript reveal distinct conformations of the capping enzymes that likely represent additional states of the capping reaction. Moreover, the structures in this manuscript contain the NELF module, which is absent in Garg et al., and reveal a "Down" conformation of the NELF module that was not observed in previously reported structures of the paused Pol II elongation complexes. The manuscript is well-written, and the new conformations of the capping/elongation factors provide new insights into transcription-coupled mRNA capping and the promoter-proximal pause release. Therefore, I recommend the publication of the manuscript in Nature Communications after the authors address the following questions.

1. The authors showed that RNGTT efficiently caps the 5' terminus of mRNA with a length of 17-20 nt but not 22-nt. These results are inconsistent with the report by Garg et al. (PMID: 37369200), in which they showed that RNGTT efficiently caps the 5' terminus of mRNA with a length longer than 22 nt. What are the possible reasons for this inconsistency? It should be discussed in the manuscript.
2. In the structure of PEC-RNGTT, the authors observed that RNGTT displaces the KOWx-KOW4 domains of SPT5. How about the other domains of SPT5? Do they bind to the same positions as in Pol II-PEC? It should be mentioned in the manuscript.
3. The cryo-EM map for the RNA in the structures of Pol II PEC-RNGTT and Pol II PEC-RNGTT-CMTR1 should be better presented. It is recommended to prepare separate figure panels for the cryo-EM map of the RNA nucleotides in the Pol II main cleft, Pol II RNA exit channel, and RNGTT.
4. The "Down" conformation of NELF is very interesting. Have the authors observed a "Down" conformation without RNGTT? In other words, is the "Down" conformation induced by RNGTT? If so, what is the implication of the RNA capping-induced conformational change of NELF? If not, is this "Down" conformation an on-pathway intermediate state of pause release? It should be discussed more in the manuscript.
5. What are the conformations of the RNA-DNA hybrid in the structures of Pol II PEC-RNGTT and Pol II PEC-RNGTT-CMTR1? In the previously reported structures of the paused elongation complex of Pol II (PMID: 30135580), the RNA-DNA hybrid adopts a tilted (or half-translocated) conformation. Does the interaction of the capping enzyme affect the DNA-RNA hybrid conformation in the active-site cleft of Pol II?

Reviewer #2 (Remarks to the Author):

In this manuscript, Le et al. describe cryo-EM structures of paused RNA polymerase II (Pol II) in complex with 5'-mRNA capping enzymes, with limited biochemical data to accompany the structures. The most glaring issue we had while reading the paper was the lack of comparison between the structures recently published by the Cramer group and the structures presented in the current work. What differences would set the current study apart to justify its publication in Nature Communications? The authors did not communicate these differences well in the current form of the manuscript. Perhaps focusing on the structural features of the paused Pol II (vs elongating Pol II in the Cramer structures) would be one direction; another could be the new position of NELF on Pol II and its importance. Our concerns are detailed below.

Major Concerns

1. This reviewer is concerned about the results of the *in vitro* guanylation assays conducted in this study. There are multiple references and statements in this manuscript that directly contradict these results.

First, there are three references cited in this manuscript that establish a minimum RNA length of ~22-24 nt for efficient capping (listed below).

i. Rasmussen, E.B., and Lis, J.T. (1993). *In vivo* transcriptional pausing and cap formation on three *Drosophila* heat shock genes. *Proc. Natl. Acad. Sci. USA* 90, 7923–7927.

<https://doi.org/10.1073/pnas.90.17.7923>.

ii. Moteki, S., and Price, D. (2002). Functional coupling of capping and transcription of mRNA. *Mol. Cell* 10, 599–609. [https://doi.org/10.1016/S1097-2765\(02\)00660-3](https://doi.org/10.1016/S1097-2765(02)00660-3).

iii. Coppola, J.A., Field, A.S., and Luse, D.S. (1983). Promoter-proximal pausing by RNA polymerase II *in vitro*: transcripts shorter than 20 nucleotides are not capped. *Proc. Natl. Acad. Sci. USA* 80, 1251–1255.

<https://doi.org/10.1073/pnas.80.5.1251>.

Second, the following two statements on lines 47-48 in the Introduction and lines 182-184 in the Results are confusing, since the authors state in line 200 that “The 5'-triphosphate of 22 nt RNA is about 48.5 Å away from the 3rd transcribed nucleotide and difficult to be accessed by the TPase of RNGTT (Fig. 2f)”

a. Lines 47-48: "It is demonstrated that capping does not occur until the transcript is elongated to 20 nt in vitro and 20-30 nt in vivo"

b. Lines 182-184: "In general, the minimum length for nascent pre-mRNA to be capped is around 20 nt, and longer products as 79 nt and 223 nt could be capped as efficiently as short transcripts."

Third, in recent work published by Garg et. al. (ref 45 in the current manuscript; <https://doi.org/10.1016/j.molcel.2023.06.002>), which reconstitutes the same complex described in this manuscript, they state that 22-nt RNA was the minimum length required for efficient capping activity (p. 2465, section: Structure of transcribing Pol II-TPase complex).

Taken together, how can the authors explain the divergence of their results from the precedents set in the field?

2. Line 143 Can the authors elaborate more on the structural features of the PEC? (e.g. translocation register, large scale conformational changes within Pol II)

Since the present study specifically chose to position Pol II at a pause, there should be some additional analysis of the paused conformation as well as how the paused state might influence Pol II interactions with the associated proteins. Also, the authors should explicitly state the pause sequence used in this study and align it with the pause consensus sequence.

3. Lines 190-197 A more rigorous analysis of these structural results is required. The process being captured structurally is a series of reactions, yet there is no mention of a strategy to "trap" the complex in a particular reaction state. Based on the data presented, it looks as though this structure captures the complex prior to cleavage of the gamma phosphate, since the RNA would likely be positioned closer to the GTase active site post cleavage.

Thus, the statement made on lines 195-197 should clearly be labeled as a speculation, since the authors do not present structural snapshots for the later steps in 5' end capping.

4. Lines 198-205 The analysis presented here is in direct contradiction with the recent structure of Pol II-RNGTT published by Garg et. al. (PDB 8P4A). For this structural analysis, a 22-nt uncapped RNA was used

and while only 19 nt could be modeled into the structure, incomplete density was observed all the way to the TPase active site.

This reviewer strongly questions the author's characterization that 22 nt RNA is inaccessible for phosphatase cleavage on lines 200-201.

The authors should at the very least add a supplementary figure illustrating a structural comparison with the structure mentioned above.

5. Garg et. al. found that for a Pol II transcribing complex prepared with a 24 nt uncapped RNA and DSIF, RNGTT, and NELF, NELF was only observed in a particle class that lacked RNGTT. The only difference between this manuscript and the work from Garg et. al. is the transcript length used. This discrepancy should also be addressed and analyzed in the manuscript. Perhaps the PEC depicted here could be the "pre capped" state, wherein NELF is prolonging the transcriptional pause and once RNGTT actually begins the series of 5' capping reactions, NELF is released from Pol II and dissociates.

6. Mandal et. al. (ref 24 in this manuscript) found using in vitro biochemical assays that capping enzymes help overcome NELF-mediated transcriptional repression. Does the paused state of Pol II structurally differ in the presence and absence of RNGTT? Can the authors provide any mechanistic insights into how RNGTT might facilitate release of Pol II from a pause? Do the structural features of the paused conformation facilitate the anchoring of RNGTT adjacent to the RNA exit channel?

7. Can the authors provide some insight into the biochemical relevance of the two observed conformations of NELF? Is there a potential connection between these states of NELF and the co-transcriptional capping activities? Can the authors offer any thoughts on how NELF affects RNGTT activity without a direct protein-protein interaction between the two on the PEC?

8. Lines 269-279 It would be helpful if the authors could add some additional details to this section. The structures they are using for comparison (8P4E and 8P4F) were assembled with a 29 nt RNA, and GTP was included during complex preparation such that the transcript would be capped and poised to enter the CMTR1 active site for methylation.

Notably, Garg et. al. could not observe density for CMTR1 when a 26 nt RNA was employed for complex preparation, so is it truly possible that CMTR1 can dock onto Pol II in advance and search for the substrate as the transcript is elongated? How can the findings be reconciled? Is there any biochemical evidence of CMTR1 binding to Pol II "in advance" in the absence of nucleic acids, RNA in particular?

9. In general, the manuscript would be strengthened by connecting the structural results obtained into a story. In its present form, it feels like disparate sets of structural observations.

Minor concerns:

There are grammatical issues and typos throughout the text that hinder clarity and readability, some examples are provided below. A more careful read through of the text should be performed to ensure appropriate verb tenses and articles are used. Additionally, the authors assume a significant pre-existing knowledge of Pol II structural components. Instead, they should label key Pol II architectural parts in an early figure (e.g., “foot”, “tip”, funnel, RNA exit channel).

Line 44: I think the authors mean to use “visualized” instead of “virtualized”.

Line 44: “complex” should be “complexes” since multiple structures are being referred to

Line 82: replace “till” with “until”

Lines 82-83: replace “get involve” with “are involved”

Line 130: “gave us a clue” is very vague phrasing

Line 212: “NLEF-E” should be “NELF-E”

Lines 45-47: “short” transcripts is very vague, the specific window for transcripts that cannot be capped should be defined more clearly.

Line 54: It is unclear what the authors mean by “GMP-capping enzyme intermediate formation”. There is a series of reactions involved in 5' end capping, so which reaction intermediate is being referred to?

Line 64: A few more sentences with background on PolII pausing would be useful, this description is a bit vague. For example, what is meant by “early pausing state” and “late pausing state”? Are the authors referring to a series of pauses encountered by RNAP in that 20-60 nt window?

Line 67: Since the authors are pointing out that 5' end capping and Pol II pausing are spatiotemporally coupled, it would be useful to illustrate a comparison between the rate of transcription by Pol II vs. the

timescale of the 5' end capping reactions. Without transcriptional pauses, is there not a sufficient time window for the series of 5' end capping reactions?

Lines 84-85: The structure of this exact complex has been published recently so it is no longer elusive as described here, can the authors instead emphasize the novel elements specific to the structure presented in this manuscript? What questions remain to be resolved that are addressed in this work?

Lines 85-90: The description of the major findings should include the broader impact of the results obtained from cryo-EM structures (e.g., how do the structural results fit into the broader context of mammalian gene expression etc...)

Lines 108-109: Were the in vitro guanylation assays performed with Pol II positioned at a pause?

Line 109 says Pol II elongation complex.

Lines 131-132: Where is the PDB reference for the DSIF model placed into the cryo-EM density?

Lines 194-197: This explanation is quite speculative. Were distinct structural states of RNGTT captured in support of this idea? Can the authors cite previous studies highlighting interdomain motions of RNGTT that are compatible with their structural data?

Figure 2 is cited before Figure 1.

It would be helpful to have a panel showing just the RNA density with the fitted model. RNA density is difficult to make out in the current EM figures. Also, show a zoomed view of the 5'-ppp on the RNA and the catalytic Cys in TPase domain.

"CE" (capping enzyme, I assume?) abbreviation never spelled out.

Reviewer #3 (Remarks to the Author):

I co-reviewed this manuscript with one of the reviewers who provided the listed reports as part of the Nature Communications initiative to facilitate training in peer review and appropriate recognition for co-reviewers.

Reviewer #4 (Remarks to the Author):

mRNA capping is an essential process in eukaryotes that increases RNA stability. In addition, efficient recruitment of mRNA to the eukaryotic translation machinery depends on the addition of a 7-methylguanylate (m7G-PPP) cap structure to the 5' end of the transcript. mRNA is capped co-transcriptionally by a dedicated complex (CCC) that is formed once the nascent transcript reaches a certain length and that contains actively transcribing RNAP and capping enzyme (CE). This process was so far not understood in all details as the only available CCC structure (*S. cerevisiae* Pol II bound to its CE, Martinez-Rucobo et al Mol. Cell 2015), provided only low resolution. In the meantime, a CCC structure from poxviral RNAP details had been published which revealed atomic details and the full trajectory of the emerging transcript reaching out to the active sites of the CE (Hillen et al., Cell 2019). Finally, this July the Cramer lab has published several atomic models of different states of the human CCC illustrating major mechanistically relevant details of the co-transcriptional capping process.

In this manuscript, Li et al. present structures of paused elongation complexes (PECs) bound to the mammalian capping enzyme RNGTT and the CMTR1 cap methyltransferase 1. These complexes were reconstituted in three steps, (1) Isolation of RNAP from porcine thymus, (2) formation of the PEC by addition of the negative elongation factors DSIF and NELF and a DNA/RNA scaffold and (3) complexation with the eukaryotic CE RNGTT and CMTR1.

The two structures are based on cryo EM reconstructions at 3.0/4.0Å resolution in which the RNGTT and CMTR1 parts are resolved significantly worse (which is expected). The structure determination appears solid and the statistics of the two model are good in relation to the limited resolution of the refinement maps.

In the light of the recent, detailed study of CCC complexes in various states by the Cramer lab the question arises, where the significance of this further study lies. Since also PECs have likewise been characterized in detail by the Cramer lab (Vos et al., Nature, 2018) and the authors state that their complex structures are similar to the PEC structures determined before, this point should be thoroughly worked out. However, I found it difficult to extract this information from the manuscript. In particular, the paragraph (starting from line 269) comparing the current structures to those from 2023 from the Cramer lab lacks a figure with a structural superposition or side-by-side comparison. There are linguistic shortcomings throughout the text that add to these difficulties. Therefore, the scientific significance of the current study appears unclear.

Before publication can be considered, those points must be addressed.

Minor points:

Articles are missing throughout the text (consider revision by a native speaker).

L44: “visualized” instead of “virtualized”

L133: This should probably read “we finally built a complete model of [...]”

L214: “Compared to” instead of “Comparing to”

L270: “that CMTR1 remains bound to the stalk” instead of “that CMTR1 remains binding stalk”

L201-202: “which is consistent with the guanylation assay results” instead of “in consistent with the guanylation assay results”

L331: “structures of eukaryotic co-transcriptional capping complexes” instead of “structures of eukaryotic co-transcriptional capping enzyme”

REVIEWER COMMENTS

Reviewer #1 (Remarks to the Author):

In this manuscript, the authors report the cryo-EM structures of the mammalian paused Pol II elongation complex bound with the elongation factors DSIF and NELF, the capping enzyme RNGTT, which catalyzes the removal of the γ -phosphate of 5'-triphosphate and the addition of GMP to the 5'-diphosphate, and the capping enzyme CMTR1, which catalyzes the 2'-OH methylation of the +1 ribonucleotide. The cryo-EM structures present clear snapshots and structural mechanisms of the critical capping event during the initial stage of mRNA synthesis. Although the recent publication by Garg et al. (PMID: 37369200) also reported similar structures of mammalian co-transcriptional capping, the structures in this manuscript reveal distinct conformations of the capping enzymes that likely represent additional states of the capping reaction. Moreover, the structures in this manuscript contain the NELF module, which is absent in Garg et al., and reveal a "Down" conformation of the NELF module that was not observed in previously reported structures of the paused Pol II elongation complexes. The manuscript is well-written, and the new conformations of the capping/elongation factors provide new insights into transcription-coupled mRNA capping and the promoter-proximal pause release. Therefore, I recommend the publication of the manuscript in Nature Communications after the authors address the following questions.

1. The authors showed that RNGTT efficiently caps the 5' terminus of mRNA with a length of 17-20 nt but not 22-nt. These results are inconsistent with the report by Garg et al. (PMID: 37369200), in which they showed that RNGTT efficiently caps the 5' terminus of mRNA with a length longer than 22 nt. What are the possible reasons for this inconsistency? It should be discussed in the manuscript.

We have examined capping activities for RNAs of various lengths from 15 nt to 33 nt. We found that RNA of 17-20 nt were more efficiently capped than others, while RNAs 16 nt, 26 nt, 28 nt and 33 nt could be weakly capped. 15 nt, 22 nt and 23 nt were hardly capped. We explained that 15 nt RNA merely reached the RNA exit and couldn't access to the activity center of RNGTT. As 22 nt and 23 nt cannot be capped in our assay, we speculated that they just crossed the activity center of TPase, and the extrusion portions of these two RNAs were unable to fold back into the activity pocket due to their intrinsic inflexibility in such a short length. When RNA gets longer, its flexibility permits it to bend, allowing the 5' triphosphate moiety to get close to the activity center again. Compared with the Garg et al.'s guanylation assay using 20 nt, 22 nt and 24 nt RNAs as substrates, 22-24 nt is required for efficient capping, in our assay it shows 17 nt is long enough for efficient capping. We discussed this in lines 191-198 in our revised manuscript.

"RNGTT could efficiently catalyze the capping reaction on RNAs from 17 to 20 nt, 16 and 21-33 nt RNAs could be slightly guanylated, but 22 and 23 nt RNAs are not (Fig. 2g). Despite the long distances between the 5' end of RNAs and the active site of Gtase, we speculated that the loop region linking the two domains of RNGTT and the sway of RNA may facilitate the enzymatic modules moving and catching the RNA substrates to complete enzymatic reactions (Fig. 2f, 2g). When it comes to 22 and 23 nt RNAs, the rigidity of the short-exposed RNA portion makes 5'-triphosphate difficult to get access to TPase activity center once 5'-end

crosses it, which may account for their failure of guanylation by RNGTT.”

2. In the structure of PEC-RNGTT, the authors observed that RNGTT displaces the KOWx-KOW4 domains of SPT5. How about the other domains of SPT5? Do they bind to the same positions as in Pol II-PEC? It should be mentioned in the manuscript.

Yes, other domains are located at similar positions to that in the Pol II-PEC structure, except that KOWx-KOW4 density is missing in our refined map. Because of the resolution limitation, we rigidly docked the model of Pol II-PEC in our map. The model of SPT5 fitted well in the map. There are no significant structural differences from SPT5 in Pol II PEC. We discussed this in lines 159-160 in our revised manuscript.

“The position of KOWx-KOW4 in PEC is occupied by T_pase domain and part of OB fold, leading to the loss of KOWx-KOW4 density (Fig. 2e), while the other parts of DSIF stay in similar position with PEC (Fig. 1b).”

3. The cryo-EM map for the RNA in the structures of Pol II PEC-RNGTT and Pol II PEC-RNGTT-CMTR1 should be better presented. It is recommended to prepare separate figure panels for the cryo-EM map of the RNA nucleotides in the Pol II main cleft, Pol II RNA exit channel, and RNGTT.

We added the new figures in Fig 2f and Supplementary Fig 4a, b, and discussed this in lines 184-187 in our revised manuscript.

“In the structure, 15 ribonucleotides could be traced in the RNA channel, and the position of the 3rd transcribed ribonucleotide is defined at RNA exit tunnel (Fig. 2d, 2f and Supplementary Fig 4a). The DNA-RNA hybrid here adopts post-translocation conformation in EC, rather than tilted conformation described in PEC (Fig. 2f and Supplementary Fig. 4b).”

4. The "Down" conformation of NELF is very interesting. Have the authors observed a "Down" conformation without RNGTT?

We didn't observe the "Down" conformation without RNGTT. NELF complex cannot be seen in the absence of RNGTT in our assembly system. We first assembled PEC with our DNA-RNA hybrid, subsequently complemented it with RNGTT. We didn't see any density for NELF in that sample, which is consistent with Garg et al.'s study¹. For we verified that NELF and RNGTT individually bound with Pol II-DSIF, we reversed the adding order of RNGTT and NELF complex to prepare the PEC-CE sample and captured the "Down" conformation as well as the "Up" conformation like PEC.

In other words, is the "Down" conformation induced by RNGTT? If so, what is the implication of the RNA capping-induced conformational change of NELF? If not, is this "Down" conformation an on-pathway intermediate state of pause release? It should be discussed more in the manuscript.

We are not sure if the "Down" conformation induced by RNGTT. We did not isolate any particles with NELF without RNGTT. We infer that it is an intermediate state of pause release and facilitates Pol II transcription²⁻⁴. We discussed this in lines 354-362 in our revised manuscript.

“Furthermore, we detected two conformations of NELF complex in the PEC-RNGTT structures. The prominent rearrangements of NELF-A-NELF-D lobe in the "Down" state result in the dissociation of NELF complex from the Pol II funnel and trigger loop, implying the transitional states of Pol II from pausing to release or other subsequent events, such as cap binding⁵. Though there is biochemical evidence indicating that capping enzymes could promote

pausing release and transition from initiation to elongation²⁻⁴, we didn't find a direct structural connection between RNGTT and NELF allosteric transition. It's still a riddle for us if RNGTT induced the new conformation of NELF."

5. What are the conformations of the RNA-DNA hybrid in the structures of Pol II PEC-RNGTT and Pol II PEC-RNGTT-CMTR1? In the previously reported structures of the paused elongation complex of Pol II (PMID: 30135580), the RNA-DNA hybrid adopts a tilted (or half-translocated) conformation. Does the interaction of the capping enzyme affect the DNA-RNA hybrid conformation in the active-site cleft of Pol II?

The DNA-RNA hybrid adopts a post-translocation conformation similar to EC, not the tiled one in PEC. We carefully analyzed the DNA-RNA hybrid model in our structures. After local refinement around the Pol II active site, densities of the last 9 ribonucleotides are observed in the Pol II cavity, pairing with the bases of the template strand, which is similar to EC structures. The DNA-RNA hybrid adopts a post-translocation conformation, not the tilted one of Pol II PEC. We discussed this in lines 184-187 in our revised manuscript.

"In the structure, 15 ribonucleotides could be traced in the RNA channel, and the position of the 3rd transcribed ribonucleotide is defined at RNA exit tunnel (Fig. 2d, 2f and Supplementary Fig 4a). The DNA-RNA hybrid here adopts post-translocation conformation in EC, rather than tilted conformation described in PEC (Fig. 2f and Supplementary Fig. 4b)."

Reviewer #2 (Remarks to the Author):

In this manuscript, Le et al. describe cryo-EM structures of paused RNA polymerase II (Pol II) in complex with 5'-mRNA capping enzymes, with limited biochemical data to accompany the structures. The most glaring issue we had while reading the paper was the lack of comparison between the structures recently published by the Cramer group and the structures presented in the current work. What differences would set the current study apart to justify its publication in Nature Communications? The authors did not communicate these differences well in the current form of the manuscript. Perhaps focusing on the structural features of the paused Pol II (vs elongating Pol II in the Cramer structures) would be one direction; another could be the new position of NELF on Pol II and its importance. Our concerns are detailed below.

Major Concerns

1. This reviewer is concerned about the results of the *in vitro* guanylation assays conducted in this study. There are multiple references and statements in this manuscript that directly contradict these results.

First, there are three references cited in this manuscript that establish a minimum RNA length of ~22-24 nt for efficient capping (listed below).

- i. Rasmussen, E.B., and Lis, J.T. (1993). *In vivo* transcriptional pausing and cap formation on three *Drosophila* heat shock genes. *Proc. Natl. Acad. Sci. USA* 90, 7923–7927. <https://doi.org/10.1073/pnas.90.17.7923>.
- ii. Moteki, S., and Price, D. (2002). Functional coupling of capping and transcription of mRNA. *Mol. Cell* 10, 599–609. [https://doi.org/10.1016/S1097-2765\(02\)00660-3](https://doi.org/10.1016/S1097-2765(02)00660-3).
- iii. Coppola, J.A., Field, A.S., and Luse, D.S. (1983). Promoter-proximal pausing by RNA polymerase II *in vitro*: transcripts shorter than 20 nucleotides are not capped. *Proc. Natl. Acad. Sci. USA* 80, 1251–1255. <https://doi.org/10.1073/pnas.80.5.1251>.

Second, the following two statements on lines 47-48 in the Introduction and lines 182-184 in the Results are confusing, since the authors state in line 200 that “The 5'-triphosphate of 22 nt RNA is about 48.5 Å away from the 3rd transcribed nucleotide and difficult to be accessed by the TPase of RNGTT (Fig. 2f)”

a. Lines 47-48: “It is demonstrated that capping does not occur until the transcript is elongated to 20 nt *in vitro* and 20-30 nt *in vivo*”

b. Lines 182-184: “In general, the minimum length for nascent pre-mRNA to be capped is around 20 nt, and longer products as 79 nt and 223 nt could be capped as efficiently as short transcripts.”

Third, in recent work published by Garg et. al. (ref 45 in the current manuscript; <https://doi.org/10.1016/j.molcel.2023.06.002>), which reconstitutes the same complex described in this manuscript, they state that 22-nt RNA was the minimum length required for

efficient capping activity (p. 2465, section: Structure of transcribing Pol II-TPase complex).

Taken together, how can the authors explain the divergence of their results from the precedents set in the field?

First question: Rasmussen et. al. used nuclei of *Drosophila* cells to set up the run-ons assay, and found that capping occurs mainly as the pre-mRNAs transcribed to 20-30 nt in length. Moteki et. al employed the purified *Drosophila* Pol II to generate an early elongation complex and never observed capping on transcripts less than 25 nt. Coppola et al. used a HeLa nuclear extract and found cap addition somewhere between 20 and 79 nt. Additionally, Hagler and Shuman used a vaccinia Pol II and found capping addition between 27 and 31 nt. Most previous studies used nuclear extract as the source of the elongation complex, which was not purified and more complicated than our recombinant complex.

Second question: We repeated the capping activity for RNAs of incongruous lengths from 15 nt to 33 nt. As shown in our revised manuscript, all RNAs could be capped except for those of 15, 22, and 23 nt. Detailed descriptions and discussion have been included in our revised manuscript in lines 191-198. As for the result in the work by Garg et. al, an upshifted band for capped 20 nt RNA could be observed in their Fig S1b, indicating the reaction is efficient.

Third question: In our *in vitro* guanylation assay, we repeatedly observed that the 22 and 23nt pre-mRNA could not be capped. We speculated the reason is that both RNAs may be just crossed the activity center of TPase, the portions outside Pol II of these two RNAs were unable to fold back into the activity pocket due to their intrinsic inflexibility in such a short length. When RNA gets longer, its flexibility permits it to bend, allowing the 5' triphosphate moiety to get close to the activity center again. As for the result in the work by Garg et. al, an upshifted band for capped 20 nt RNA could be observed in their Fig S1b, indicating the reaction is efficient. We discussed about this in lines 191-198 in our revised manuscript.

[Redacted]

Please see Figure S1 in Garg G, et al. Structural insights into human co-transcriptional capping. Mol Cell 83, 2464-2477 e2465806 (2023). 10.1016/j.molcel.2023.06.002

“RNGTT could efficiently catalyze the capping reaction on RNAs from 17 to 20 nt, 16 and 21-

33 nt RNAs could be slightly guanylated, but 22 and 23 nt RNAs are not (Fig. 2g). Despite the long distances between the 5' end of RNAs and the active site of Gtase, we speculated that the loop region linking the two domains of RNGTT and the sway of RNA may facilitate the enzymatic modules moving and catching the RNA substrates to complete enzymatic reactions (Fig. 2f, 2g). When it comes to 22 and 23 nt RNAs, the rigidity of the short-exposed RNA portion makes 5'-triphosphate difficult to get access to TPase activity center once 5'-end crosses it, which may account for their failure of guanylation by RNGTT."

2. Line 143 Can the authors elaborate more on the structural features of the PEC? (e.g. translocation register, large scale conformational changes within Pol II)

Since the present study specifically chose to position Pol II at a pause, there should be some additional analysis of the paused conformation as well as how the paused state might influence Pol II interactions with the associated proteins.

We added this part in lines 159-160, 184-187 and 221-224 in our revised manuscript.

"The position of KOWx-KOW4 in PEC is occupied by Tase domain and part of OB fold, leading to the loss of KOWx-KOW4 density (Fig. 2e), while the other parts of DSIF stay in similar position with PEC (Fig. 1b)."

"In the structure, 15 ribonucleotides could be traced in the RNA channel, and the position of the 3rd transcribed ribonucleotide is defined at RNA exit tunnel (Fig. 2d, 2f and Supplementary Fig 4a). The DNA-RNA hybrid here adopts post-translocation conformation in EC, rather than tilted conformation described in PEC (Fig. 2f and Supplementary Fig. 4b)."

"The C-terminal region of NELF-D detaches from RPB1 funnel and trigger loop, leaving the funnel partially open and convenient for NTP delivery to the active site (Fig. 3a and Supplementary Fig. 5a). In addition, trigger loop swings away from funnel helices, recovering its mobility under this conformation. (Fig. 3g)"

Also, the authors should explicitly state the pause sequence used in this study and align it with the pause consensus sequence.

The sequence used in this study is an elongation sequence used in an activated transcription complex (Pol II-DSIF-SPT6-PAF1 complex), rather than a pause sequence. Therefore, we didn't align it with the pause consensus sequence here. We didn't intend to confine the capping event at the pausing stage at first. Therefore, we chose the elongation sequence to test various transcription complexes, including elongation complex, Pol II-DSIF elongation complex, paused elongation complex, and activated transcription complex, to figure out which factors capping enzymes could co-exist with. In our gradient sedimentation assay, it is shown that RNGTT binding is not dependent on the paused state, but phosphorylated Pol II and DSIF are very helpful.

3. Lines 190-197 A more rigorous analysis of these structural results is required. The process being captured structurally is a series of reactions, yet there is no mention of a strategy to "trap" the complex in a particular reaction state. Based on the data presented, it looks as though this structure captures the complex prior to cleavage of the gamma phosphate, since the RNA would likely be positioned closer to the GTase active site post cleavage.

Thus, the statement made on lines 195-197 should clearly be labeled as a speculation, since the authors do not present structural snapshots for the later steps in 5' end capping.

We reorganized this part in our revised manuscript. The capping of pre-mRNA consists of two steps: the dephosphorylation and guanylation of 5' triphosphate. We didn't aim at

catching a specific reaction state of the capping enzyme in complex with Pol II. We here inhibited the guanylation by adding GMPPNP to prevent the capping enzyme from dissociating from the Pol II upon the two steps were accomplished. We also anticipated separating the different conformations of RNA caught by each domain. However, we only separated one class of RNA conformation, in which the RNA is guided to the TPase domain. Detailed descriptions and discussion have been included in our revised manuscript in lines 193-198.

“Despite the long distances between the 5' end of RNAs and the active site of GTase, we speculated that the loop region linking the two domains of RNGTT and the sway of RNA may facilitate the enzymatic modules moving and catching the RNA substrates to complete enzymatic reactions (Fig. 2f, 2g). When it comes to 22 and 23 nt RNAs, the rigidity of the short-exposed RNA portion makes 5'-triphosphate difficult to get access to TPase activity center once 5'-end crosses it, which may account for their failure of guanylation by RNGTT.”

4. Lines 198-205 The analysis presented here is in direct contradiction with the recent structure of Pol II-RNGTT published by Garg et. al. (PDB 8P4A). For this structural analysis, a 22-nt uncapped RNA was used and while only 19 nt could be modeled into the structure, incomplete density was observed all the way to the TPase active site.

This reviewer strongly questions the author's characterization that 22 nt RNA is inaccessible for phosphatase cleavage on lines 200-201.

We reorganized this part in our revised manuscript in lines 193-198.

“Despite the long distances between the 5' end of RNAs and the active site of GTase, we speculated that the loop region linking the two domains of RNGTT and the sway of RNA may facilitate the enzymatic modules moving and catching the RNA substrates to complete enzymatic reactions (Fig. 2f, 2g). When it comes to 22 and 23 nt RNAs, the rigidity of the short-exposed RNA portion makes 5'-triphosphate difficult to get access to TPase activity center once 5'-end crosses it, which may account for their failure of guanylation by RNGTT.”

The authors should at the very least add a supplementary figure illustrating a structural comparison with the structure mentioned above.

We added the structural comparison in Supplemental Fig. 6 and lines 238-257. Our first edition didn't contain this part because the models of Pol II-RNGTT (PDB 8P4A,8P4B) and Pol II-DSIF-RNGTT (PDB 8P4C,8P4D) published by Garg et. al. didn't release by the end of our submission.

“Structural comparison with transcribing Pol II-DSIF-RNGTT complex

When we prepared for this manuscript, a study resolving structures of co-transcriptional capping enzymes was published ⁶. They assembled RNGTT on the transcribing Pol II and Pol II-DSIF-NELF complex using longer RNAs respectively. For the latter complex, we both used the same components to assemble the PEC-RNGTT complex, except that they used CDK7 to phosphorylate Pol II and added GTP as a substrate for guanylation. Unexpectedly, they didn't acquire particles containing both RNGTT and NELF. They got two conformations of RNGTT on transcribing Pol II-DSIF complex. One is intact with RNA pointing towards the TPase

domain (PDB ID:8P4C), and the other lacks visible density of the TPase domain with RNA pointing towards the GTase domain (PDB ID:8P4D) (Supplementary Fig. 6b and 6c). The TPase in both studies is similarly located around the RNA exit tunnel and GTase anchors to Pol II stalk through its OB loop (Supplementary Fig. 6a-c). They displayed how KOWx-KOW4 of SPT5 interacts with TPase domain. KOWx-KOW4 as well as KOW1-L1 shows conformational changes that are not observed in our map (Supplementary Fig. 6d-f). Compared with the PEC-RNGTT structure, GTase domain in the TPase-invisible structure could swing closer to the RNA exit tunnel (Supplementary Fig. 6a and 6c). We speculate the swing is associated with functional status of GTase domain, as GMPPNP we used in assembly can theoretically lock its active center. Besides, the analog keeps the capping reaction in unfinished status, which may help us capture the NELF-containing capping complex. Taken together, our PEC-RNGTT complex seems to be in an earlier stage of the capping process.”

5. Garg et. al. found that for a Pol II transcribing complex prepared with a 24 nt uncapped RNA and DSIF, RNGTT, and NELF, NELF was only observed in a particle class that lacked RNGTT. The only difference between this manuscript and the work from Garg et. al. is the transcript length used. This discrepancy should also be addressed and analyzed in the manuscript.

In our opinion, the length of RNA is not the main reason for NELF observation. The strategy of the complex assembling mattered. At first, we assembled the complex in the order of phosphorylated Pol II, DNA-RNA hybrid, non-template strand DNA, DSIF, NELF, RNGTT, we obtained the same result as Garg et. al. Referring to the references and our gradient sedimentation assay, we speculated RNGTT is more dependent on the CTD phosphorylation and SPT5. We swapped NELF and RNGTT, and acquired the two conformations of Pol II-PEC-Capping enzyme.

Perhaps the PEC depicted here could be the “pre capped” state, wherein NELF is prolonging the transcriptional pause and once RNGTT actually begins the series of 5' capping reactions, NELF is released from Pol II and dissociates.

We agree that it may be the “pre capped” state, at least non-guanylated. We can infer that once RNGTT actually completes guanylation, pausing will be released verified by the conformation change of NELF-AC module in “Down” conformation, but NELF may not dissociate from Pol II for the following steps like CBC binding.

6. Mandal et. al. (ref 24 in this manuscript) found using in vitro biochemical assays that capping enzymes help overcome NELF-mediated transcriptional repression. Does the paused state of Pol II structurally differ in the presence and absence of RNGTT? Can the authors provide any mechanistic insights into how RNGTT might facilitate release of Pol II from a pause?

Yes, there is an obvious conformational change in the cleft of Pol II, though the overall structure is quite similar with PEC. In the presence of RNGTT, the NELF-C/D in the “down” conformation was away from the funnel region, abolishing the interaction with the trigger loop, for which the trigger loop adopted another conformation. The binding of RNGTT

probably facilitates Pol II's mobility and recovers the tilted RNA to a normal angle. We discussed this in lines 184-187 and 221-224 in our revised manuscript.

"In the structure, 15 ribonucleotides could be traced in the RNA channel, and the position of the 3rd transcribed ribonucleotide is defined at RNA exit tunnel (Fig. 2d, 2f and Supplementary Fig 4a). The DNA-RNA hybrid here adopts post-translocation conformation in EC, rather than tilted conformation described in PEC (Fig. 2f and Supplementary Fig. 4b)."

"The C-terminal region of NELF-D detaches from RPB1 funnel and trigger loop, leaving the funnel partially open and convenient for NTP delivery to the active site (Fig. 3a and Supplementary Fig. 5a). In addition, trigger loop swings away from funnel helices, recovering its mobility under this conformation. (Fig. 3g)"

Do the structural features of the paused conformation facilitate the anchoring of RNGTT adjacent to the RNA exit channel?

We are not sure about the help of the paused conformation to RNGTT docking. The structures in Pol II-PEC and this study both missed the majority of information of the NELF-E component, which is considered to bind with RNA and plays an important role in the Pol II pausing. The rest components of the NELF complex are far away from the RNA exit channel and the stalk and do not contribute to RNGTT docking to the position. The other evidence is in our gradient sedimentation assay, the NELF complex did not promote RNGTT interacting with Pol II-DSIF.

7. Can the authors provide some insight into the biochemical relevance of the two observed conformations of NELF?

We thought the "Up" conformation is a pausing status while the "Down" conformation is a pausing released status. We observed two conformations of NELF complex in the presence of RNGTT, the "Up" conformation is the same as that of Pol II-PEC. The "Up" conformation represents the paused Pol II due to Pol II mobility constraint and DNA-RNA tilting. The "Down" conformation in this paper is a pausing released status of Pol II-NELF complex, allowing the RNA transcription goes on²⁻⁴.

Is there a potential connection between these states of NELF and the co-transcriptional capping activities? Can the authors offer any thoughts on how NELF affects RNGTT activity without a direct protein-protein interaction between the two on the PEC?

We thought there is no connection between NELF states and capping activities. We did not observe any direct interaction between NELF and RNGTT and addition of NELF in guanylation assay didn't affect the reaction.

8. Lines 269-279 It would be helpful if the authors could add some additional details to this section. The structures they are using for comparison (8P4E and 8P4F) were assembled with a 29 nt RNA, and GTP was included during complex preparation such that the transcript would be capped and poised to enter the CMTR1 active site for methylation.

We discussed this in lines 360-366 in our revised manuscript.

“For the PEC-RNGTT-CMTR1, we captured both full-length RNGTT and CMTR1 visibly arrayed adjacent to the Pol II stalk. The catalytic domain of CMTR1 is further away from the RNA exit tunnel than latest two structures of transcribing Pol II with CMTR1(PDB ID:8P4E and 8P4F) 45, probably because the longer RNA we used is difficult for CMTR1 to catch and hinder us not to find the RNA density, implying that our model without RNA is perhaps an initial position for CMTR1 docking on the Pol II prior to substrate recognition.”

Notably, Garg et. al. could not observe density for CMTR1 when a 26 nt RNA was employed for complex preparation, so is it truly possible that CMTR1 can dock onto Pol II in advance and search for the substrate as the transcript is elongated? How can the findings be reconciled? Is there any biochemical evidence of CMTR1 binding to Pol II “in advance” in the absence of nucleic acids, RNA in particular?

We have examined if CMTR1 could bind with Pol II with shorter RNA, 20nt, or even without RNA by glycerol gradient centrifugation. The results showed that the association between CMTR1 and PEC was independent of RNA length which testified to our speculation. For Pol II without DNA is non-functional and not the related status in this study we didn't test the interaction of CMTR1 with Pol II without DNA.

9. In general, the manuscript would be strengthened by connecting the structural results

obtained into a story. In its present form, it feels like disparate sets of structural observations. We managed to reorganize the contents and connect all the structures into a story in our revised manuscript,

Minor concerns:

There are grammatical issues and typos throughout the text that hinder clarity and readability, some examples are provided below. A more careful read through of the text should be performed to ensure appropriate verb tenses and articles are used. Additionally, the authors assume a significant pre-existing knowledge of Pol II structural components. Instead, they should label key Pol II architectural parts in an early figure (e.g., “foot”, “tip”, funnel, RNA exit channel).

Line 44: I think the authors mean to use “visualized” instead of “virtualized”.

Line 44: “complex” should be “complexes” since multiple structures are being referred to

Line 82: replace “till” with “until”

Lines 82-83: replace “get involve” with “are involved”

Line 130: “gave us a clue” is very vague phrasing

Line 212: “NLEF-E” should be “NELF-E”

Lines 45-47: “short” transcripts is very vague, the specific window for transcripts that cannot be capped should be defined more clearly.

All minor points have been revised in the manuscript.

Line 54: It is unclear what the authors mean by “GMP-capping enzyme intermediate formation”. There is a series of reactions involved in 5' end capping, so which reaction intermediate is being referred to?

It is referred to the intermediate of GMP-GTase intermediate for guanylation.

Line 64: A few more sentences with background on PolIII pausing would be useful, this description is a bit vague. For example, what is meant by “early pausing state” and “late pausing state”? Are the authors referring to a series of pauses encountered by RNAP in that 20-60 nt window?

Tome et. al. identified two pause classes using single-molecule nascent RNA sequencing⁷. One population from 20-32nt (early) and another from 32-60nt (late). Early- and late-pause transcription starting nucleotides have different capped and uncapped distributions, reflecting the interplay of capping and Pol II pausing.

Line 67: Since the authors are pointing out that 5' end capping and Pol II pausing are spatiotemporally coupled, it would be useful to illustrate a comparison between the rate of transcription by Pol II vs. the timescale of the 5' end capping reactions. Without transcriptional pauses, is there not a sufficient time window for the series of 5' end capping reactions?

RNA Pol II velocity has been shown to control many co-transcriptional processes, such as constitutive splicing, alternative splicing, alternative polyadenylation, and transcription termination, probably including 5' end capping. It's reported that the transcription speed of Pol II is about 0.5kb/min in the first 10-15kb, that is 8bp/s⁸. Nevertheless, the steady-state turnover number for the triphosphatase domain in MCE1 is about 1–2 molecules of Pi release

per second per enzyme⁹. Consequently, transcription speed is required to slow down for co-transcription processes, or the enzymes are ready to recognize the binding sites upon the sites appear in the transcripts. This is the possible reason why Pol II-PEC and capping enzyme are spatiotemporally correlated.

Lines 84-85: The structure of this exact complex has been published recently so it is no longer elusive as described here, can the authors instead emphasize the novel elements specific to the structure presented in this manuscript? What questions remain to be resolved that are addressed in this work?

We have addressed this study is to solve the structures of capping enzyme coupled with paused elongation complex in lines 88-95 and 375-378 in our revised manuscript.

“In PEC-RNGTT, we unveil that RNGTT is docked adjacent to the Pol II stalk with the OB fold inserting into the root of stalk, and positions TPase at the RNA exit tunnel. Besides, we observe a new conformation of NELF in the complex, which implies that the binding of capping enzyme relieves the NELF-mediated Pol II pausing. In PEC-RNGTT-CMTR1, we represent the RNGTT and CMTR1 array at the Pol II periphery bound with different interfaces of the Pol II stalk, neighboring the RNA exit tunnel. We show the structural evidence that capping enzymes coexist with pausing factors, which are helpful in understanding the correlation of 5' end capping modifications and Pol II transcription processes.”

“Taken together, our cryo-EM structures shed light on the co-transcriptional pre-mRNA capping and methylation to some extent, particularly correlating capping with Pol II paused elongation complex and capturing several conformational changes of enzymes and transcription factors.”

Lines 85-90: The description of the major findings should include the broader impact of the results obtained from cryo-EM structures (e.g., how do the structural results fit into the broader context of mammalian gene expression etc..)

We broaden our structures' significance in lines 88-95 and 375-378 in our revised manuscript.

Lines 108-109: Were the in vitro guanylation assays performed with Pol II positioned at a pause?

Line 109 says Pol II elongation complex.

Based on the gradient sedimentation results, NELF complex has no effect on RNGTT binding, and Pol II-DSIF is sufficient. Therefore, we employed Pol II-DSIF complex to carry out the guanylation assays.

Lines 131-132: Where is the PDB reference for the DSIF model placed into the cryo-EM density?

We added the PDB reference for DSIF model in lines 131-133 in our revised manuscript.

“We positioned the models of TPase of RNGTT (PDB ID:1I9S), GTase of RNGTT (PDB ID:3S24), DSIF (PDB ID:6GML) and NELF moiety (PDB ID:6GML) into corresponding densities and adjusted the model in Coot.”

Lines 194-197: This explanation is quite speculative. Were distinct structural states of RNGTT captured in support of this idea? Can the authors cite previous studies highlighting

interdomain motions of RNGTT that are compatible with their structural data?

When we submitted this paper, the structures were not released from the databank. Compared with the transcribing Pol II-DSIF structures in previous study, the positions of TPase and GTase varies among the three models. The GTase domain gradually gets close to the RNA exit tunnel to catalyze the RNA substrate, in the meantime, the TPase alters its conformation until it cannot be reconstructed. We complemented it as the supplementary figures in the revised manuscript.

Figure 2 is cited before Figure 1.

We have changed it in the revised manuscript.

It would helpful to have a panel showing just the RNA density with the fitted model. RNA density is difficult to make out in the current EM figures. Also, show a zoomed view of the 5'-ppp on the RNA and the catalytic Cys in TPase domain.

We have added a figure to show the RNA density and highlighted the 5'-ppp on the RNA and the catalytic Cys in TPase domain in the revised manuscript.

“CE” (capping enzyme, I assume?) abbreviation never spelled out.

We have added it in the revised manuscript.

Reviewer #3 (Remarks to the Author):

I co-reviewed this manuscript with one of the reviewers who provided the listed reports as part of the Nature Communications initiative to facilitate training in peer review and appropriate recognition for co-reviewers.

Reviewer #4 (Remarks to the Author):

mRNA capping is an essential process in eukaryotes that increases RNA stability. In addition, efficient recruitment of mRNA to the eukaryotic translation machinery depends on the addition of a 7-methylguanylate (m7G-PPP) cap structure to the 5' end of the transcript. mRNA is capped co-transcriptionally by a dedicated complex (CCC) that is formed once the nascent transcript reaches a certain length and that contains actively transcribing RNAP and capping enzyme (CE). This process was so far not understood in all details as the only available CCC structure (*S. cerevisiae* Pol II bound to its CE, Martinez-Rucobo et al Mol. Cell 2015), provided only low resolution. In the meantime, a CCC structure from poxviral RNAP details had been published which revealed atomic details and the full trajectory of the emerging transcript reaching out to the active sites of the CE (Hillen et al., Cell 2019). Finally, this July the Cramer lab has published several atomic models of different states of the human CCC illustrating major mechanistically relevant details of the co-transcriptional capping process.

In this manuscript, Li et al. present structures of paused elongation complexes (PECs) bound to the mammalian capping enzyme RNGTT and the CMTR1 cap methyltransferase 1. These complexes were reconstituted in three steps, (1) Isolation of RNAP from porcine thymus, (2) formation of the PEC by addition of the negative elongation factors DSIF and NELF and a DNA/RNA scaffold and (3) complexation with the eukaryotic CE RNGTT and CMTR1.

The two structures are based on cryo EM reconstructions at 3.0/4.0Å resolution in which the RNGTT and CMTR1 parts are resolved significantly worse (which is expected). The structure determination appears solid and the statistics of the two model are good in relation to the limited resolution of the refinement maps.

In the light of the recent, detailed study of CCC complexes in various states by the Cramer lab the question arises, where the significance of this further study lies.

We complemented with the content in lines 88-95 and 375-378 in our revised manuscript.

"In PEC-RNGTT, we unveil that RNGTT is docked adjacent to the Pol II stalk with the OB fold inserting into the root of stalk, and positions TPase at the RNA exit tunnel. Besides, we observe a new conformation of NELF in the complex, which implies that the binding of capping enzyme relieves the NELF-mediated Pol II pausing. In PEC-RNGTT-CMTR1, we represent the RNGTT and CMTR1 array at the Pol II periphery bound with different interfaces of the Pol II stalk, neighboring the RNA exit tunnel. We show the structural evidence that capping enzymes coexist with pausing factors, which are helpful in understanding the correlation of 5' end capping modifications and Pol II transcription processes."

"Taken together, our cryo-EM structures shed light on the co-transcriptional pre-mRNA capping and methylation to some extent, particularly correlating capping with Pol II paused elongation complex and capturing several conformational changes of enzymes and transcription factors."

Since also PECs have likewise been characterized in detail by the Cramer lab (Vos et al., Nature, 2018) and the authors state that their complex structures are similar to the PEC structures determined before, this point should be thoroughly worked out. However, I found it difficult to extract this information from the manuscript.

We reorganized the content 143-146, 159-160, 184-187 and 221-224 in our revised manuscript.

“The structure of PEC-RNGTT shows that Pol II is similar to the previously determined in PEC structure¹⁰. DSIF domains wrap around the Pol II body from the upstream DNA entry to nascent RNA exit tunnel. NELF complex hangs at the periphery of the Pol II body (Fig. 1b, c and Supplementary Fig. 3a), which will be discussed later.”

“The position of KOWx-KOW4 in PEC is occupied by Tase domain and part of OB fold, leading to the loss of KOWx-KOW4 density (Fig. 2e), while the other parts of DSIF stay in similar position with PEC (Fig. 1b).”

“In the structure, 15 ribonucleotides could be traced in the RNA channel, and the position of the 3rd transcribed ribonucleotide is defined at RNA exit tunnel (Fig. 2d, 2f and Supplementary Fig 4a). The DNA–RNA hybrid here adopts post-translocation conformation in EC, rather than tilted conformation described in PEC (Fig. 2f and Supplementary Fig. 4b).”

“The C-terminal region of NELF-D detaches from RPB1 funnel and trigger loop, leaving the funnel partially open and convenient for NTP delivery to the active site (Fig. 3a and Supplementary Fig. 5a). In addition, trigger loop swings away from funnel helices, recovering its mobility under this conformation. (Fig. 3g)”

In particular, the paragraph (starting from line 269) comparing the current structures to those from 2023 from the Cramer lab lacks a figure with a structural superposition or side-by-side comparison.

It is already organized in the paper of the original manuscript. We used a side-by-side format to compare in Fig. 4 and Supplementary Fig. 6.

There are linguistic shortcomings throughout the text that add to these difficulties. Therefore, the scientific significance of the current study appears unclear.

We complemented with the content in lines 375-378 in our revised manuscript.

“Taken together, our cryo-EM structures shed light on the co-transcriptional pre-mRNA capping and methylation to some extent, particularly correlating capping with Pol II paused elongation complex and capturing several conformational changes of enzymes and transcription factors.”

Before publication can be considered, those points must be addressed.

Minor points:

Articles are missing throughout the text (consider revision by a native speaker).

L44: “visualized” instead of “virtualized”

L133: This should probably read “we finally built a complete model of [...]”

L214: “Compared to” instead of “Comparing to”

L270: “that CMTR1 remains bound to the stalk” instead of “that CMTR1 remains binding stalk”

L201-202: “which is consistent with the guanylation assay results” instead of “in consistent with the guanylation assay results”

L331: “structures of eukaryotic co-transcriptional capping complexes” instead of “structures of eukaryotic co-transcriptional capping enzyme”

All these minor points have been revised in the manuscript.

- 1 Bernecky, C., Plitzko, J. M. & Cramer, P. Structure of a transcribing RNA polymerase II–DSIF complex reveals a multidentate DNA–RNA clamp. *Nature Structural & Molecular Biology* **24**, 809–815, doi:10.1038/nsmb.3465 (2017).
- 2 Mandal, S. S. *et al.* Functional interactions of RNA-capping enzyme with factors that positively and negatively regulate promoter escape by RNA polymerase II. *Proceedings of the National Academy of Sciences* **101**, 7572–7577, doi:10.1073/pnas.0401493101 (2004).
- 3 Kim, H. J. *et al.* mRNA capping enzyme activity is coupled to an early transcription elongation. *Mol Cell Biol* **24**, 6184–6193, doi:Doi 10.1128/Mcb.24.14.6184–6193.2004 (2004).
- 4 Fujiwara, R., Damodaren, N., Wilusz, J. E. & Murakami, K. The capping enzyme facilitates promoter escape and assembly of a follow-on preinitiation complex for reinitiation. *Proc Natl Acad Sci U S A* **116**, 22573–22582, doi:10.1073/pnas.1905449116 (2019).
- 5 Aoi, Y. *et al.* NELF Regulates a Promoter-Proximal Step Distinct from RNA Pol II Pause-Release. *Mol Cell* **78**, 261–274 e265, doi:10.1016/j.molcel.2020.02.014 (2020).
- 6 Garg, G. *et al.* Structural insights into human co-transcriptional capping. *Mol Cell* **83**, 2464–+, doi:10.1016/j.molcel.2023.06.002 (2023).
- 7 Tome, J. M., Tippens, N. D. & Lis, J. T. Single-molecule nascent RNA sequencing identifies regulatory domain architecture at promoters and enhancers. *Nature Genetics* **50**, 1533–1541, doi:10.1038/s41588-018-0234-5 (2018).
- 8 Muniz, L., Nicolas, E. & Trouche, D. RNA polymerase II speed: a key player in controlling and adapting transcriptome composition. *Embo J* **40**, doi:ARTN e105740 10.15252/emboj.2020105740 (2021).
- 9 Ho, C. K. *et al.* The guanylyltransferase domain of mammalian mRNA capping enzyme binds to the phosphorylated carboxyl-terminal domain of RNA polymerase II. *J Biol Chem* **273**, 9577–9585, doi:DOI 10.1074/jbc.273.16.9577 (1998).
- 10 Vos, S. M., Farnung, L., Urlaub, H. & Cramer, P. Structure of paused transcription complex Pol II–DSIF–NELF. *Nature* **560**, 601–606, doi:10.1038/s41586-018-0442-2 (2018).

REVIEWER COMMENTS

Reviewer #1 (Remarks to the Author):

The authors have adequately addressed the reviews, and the manuscript is suitable for publication.

Line 769-770, 'probably because the longer RNA we used is difficult for CMTR1 to catch and hinder us not to find the RNA density', 'not' should be removed.

Reviewer #2 (Remarks to the Author):

Point 2:

The authors' response, i.e., that the nucleic acid sequence used to reconstitute transcription complexes for structure determination was that of an elongation/transcribing complex, made us realize that there is a disconnect between what the authors define as a "paused" RNA polymerase and what paused RNAP actually is. It appears that the authors' definition of Pol II paused state is simple association of NELF with Pol II. However, a paused RNAP should itself display key properties, based on PEC structures determined to date: a tilted DNA-RNA hybrid (with the template DNA base unable to base-pair with an incoming NTP substrate) and swiveling of several structural components of RNAP.

With the above in mind, how do the authors know that Pol II in their structures is actually in a paused state? They say that DNA-RNA hybrid is post-translocated, not tilted. They don't comment on the swiveling status of Pol II. They don't perform nucleotide addition assay showing that their Pol II adds the next NTP much more slowly in the presence of NELF than in its absence. This is critical to address because the entire point of the paper (and what distinguishes it from Garg et al), the way we understand it, is that capping enzymes are able to associate with paused Pol II; therefore, paused Pol II is an important transcription entity for mRNA processing.

The added structural description of what the authors define as a PEC is not informative, beyond the DNA-RNA hybrid conformation.

Point 4:

We appreciate the authors adding the supplemental figure showing structure comparison with Garg et al.

Added “Structural comparison” section: the authors state that Garg et al. added GTP to their complexes, but we can’t find that information in the published Garg et al. No GTP addition is mentioned in Garg et al. sample preparation protocol for this particular complex.

Point 5:

The spelt-out difference in the order of component addition for cryo-EM sample preparation was helpful. Based on this order description, RNGTT was added before NELF. BUT doesn't this contradict the statement/conclusion that paused Pol II promotes capping? Meaning, don't you need NELF to first bind Pol II to get it into what the authors assume is a paused state?

Points 6 and 7:

In the revised text, authors added statements like “Our structures unravel that capping enzymes RNGTT and CMTR1 can directly bind the paused elongation complex, and shed light on how pre-mRNA 5'-end capping couples with transcription at the pausing stage” (Abstract) and “We show the structural evidence that capping enzymes coexist with pausing factors, which are helpful in understanding the correlation of 5' end capping modifications and Pol II transcription processes” (lines 93-94). Yet, in the response they say that there is no connection between capping activities and NELF states (up/down, paused/unpaused Pol II... although the “up” NELF is yet to be biochemically linked to paused Pol II) or NELF/capping enzyme association with Pol II.

This brings us back to novelty of the current study: what new is learned? As we see it, what distinguishes this current study from Garg et al. is that here the authors captured NELF in complex with Pol II and capping enzymes (RNGTT). But how exactly does this add to the mechanistic understanding of the link between transcriptional pausing and mRNA capping?

Point 9:

With all due respect, the writing should still be significantly improved, including that in the added text. We strongly recommend seeking assistance of a writing center or such resource to bring writing/storytelling quality up to Nat Commun level.

We were satisfied with the authors’ responses to the rest of our original review. We appreciate the authors performing additional experiments, e.g., guanylation with intermediate RNA lengths, and adding “big picture”/conclusion statements throughout.

Reviewer #3 (Remarks to the Author):

Reviewer #4 (Remarks to the Author):

Within the revised manuscript the authors now provide a much-improved comparison of their PEC-RNGTT-CMTR1 structure to transcribing Pol II-DSIF-RNGTT-CMTR1 and Pol II-DSIF-CMTR1. Also, most other points have been addressed reasonably now. In particular, the discrepancy in optimum RNA length in the capping assay, which was a concern raised by reviewers #1 and #2 is now addressed and discussed based on the improved structural comparison shown in Fig. S6 that now also visualizes the RNA trajectory to the CE. All in all, important aspects emerge in from the manuscript that set this study apart from the Cramer lab's paper published last year ("Down" conformation of NELF, different capping length optimum). It is up to the editor to decide whether this is significant enough to justify publication in Nature communications.

Further points:

Fig. 4c/d is never referenced in the text.

REVIEWER COMMENTS

Reviewer #1 (Remarks to the Author):

The authors have adequately addressed the reviews, and the manuscript is suitable for publication.

Line 769-770, 'probably because the longer RNA we used is difficult for CMTR1 to catch and hinder us not to find the RNA density', 'not' should be removed.

Thanks for the correction. We have modified the sentence in the revised manuscript:

“probably because the longer RNA we used is difficult for CMTR1 to catch and hinder us to find the RNA density.”

Reviewer #2 (Remarks to the Author):

Point 2:

The authors' response, i.e., that the nucleic acid sequence used to reconstitute transcription complexes for structure determination was that of an elongation/transcribing complex, made us realize that there is a disconnect between what the authors define as a “paused” RNA polymerase and what paused RNAP actually is. It appears that the authors' definition of Pol II paused state is simple association of NELF with Pol II. However, a paused RNAP should itself display key properties, based on PEC structures determined to date: a tilted DNA-RNA hybrid (with the template DNA base unable to base-pair with an incoming NTP substrate) and swiveling of several structural components of RNAP.

With the above in mind, how do the authors know that Pol II in their structures is actually in a paused state? They say that DNA-RNA hybrid is post-translocated, not tilted. They don't comment on the swiveling status of Pol II. They don't perform nucleotide addition assay showing that their Pol II adds the next NTP much more slowly in the presence of NELF than in its absence. This is critical to address because the entire point of the paper (and what distinguishes it from Garg et al), the way we understand it, is that capping enzymes are able to associate with paused Pol II; therefore, paused Pol II is an important transcription entity for mRNA processing.

The added structural description of what the authors define as a PEC is not informative, beyond the DNA-RNA hybrid conformation.

We compared the Pol II structure of PEC-RNGTT with those of Pol II-DSIF EC and PEC structures. The core and shelf modules adopt similar conformation. It's also reported that the core and shelf modules keep the positions between Pol II-DSIF EC and PEC^[1,2]. The swiveling of core and shelf modules observed in reactivation from an arrested state in yeast Pol II, are not found in the reported PEC and our structures. We supplemented the related comments in the results and discussion section.

Line 235-237 “In addition, the core module, containing RPB8 and RPB1 funnel, and the shelf module, containing RPB1 cleft and foot domains, of PEC-RNGTT structure adopt a conformation similar to those of Pol II-DSIF and PEC structures.”

Line 367-372 “Compared PEC-RNGTT with Pol II-DSIF EC and PEC structures, there is no conformational change of the core and shelf modules among them, which is also reported in PEC structure^{16,43}. The swiveling of core module and shelf modules observed in reactivation from an arrested state in yeast Pol II, are not found in the reported PEC and our structures. We inferred that the disappearance of swiveling possibly results from the different factors and mechanisms involved in the pause-release and arrested-reactivation processes.”

We have performed nucleotide addition assay with a FAM-labelled 17-nt RNA to demonstrate our RNAP is indeed paused, in which the template and non-template DNA sequence were used in our cryo-EM samples. We found that the transcription rate of Pol II is slower in the presence of NELF than its absence, which is repeatedly shown in the assay for point 6 and 7. Besides, RNGTT itself has no obvious influence on transcription efficiency. These results demonstrated that RNAP with the RNA and DNA duplex in the cryo-EM samples could be paused mediated by NELF.

Point 4:

We appreciate the authors adding the supplemental figure showing structure comparison with Garg et al.

Added “Structural comparison” section: the authors state that Garg et al. added GTP to their complexes, but we can’t find that information in the published Garg et al. No GTP addition is mentioned in Garg et al. sample preparation protocol for this particular complex.

We checked the method details section of the publication by Garg et al. They described it as follows:

“All following concentrations refer to final concentrations in 100 μ L complex formation reaction. Pol II (1 μ M) and hybrid (1 μ M) were incubated for 15 min on ice. Non-template DNA (5’-CCA TTG AGA GCG GCC CTT GTG TTC AGG AGC CAG CAG GGA GCT GGG AGC-3’) (1.5 μ M) was added, and the sample incubated for 10 min on ice. CDK7 (0.5 μ M) and GTP (1mM) were added, and the sample further incubated for 5 min at 30 $^{\circ}$ C.

To assemble the Pol II-RNGTT complexes, RNGTT (C126S/WT/dOB) (3 μ M) was added. To assemble the Pol II-DSIF-RNGTT complex, DSIF (1.5 μ M), RNGTT (3 μ M) and NELF (4 μ M) were added. To assemble the Pol II-DSIF-RNGTT-CMTR1, DSIF (1.5 μ M), RNGTT (3 μ M), CMTR1 (3 μ M), and NELF (4 μ M) were added. Finally, water, and a 10-fold compensation buffer were added to reach the final buffer conditions, and the sample further incubated for 30 min at 30 $^{\circ}$ C.”

We thought that GTP as well as CDK7 was added in all the samples, The authors only mentioned the concentrations of proteins in the second paragraph and omitted other components and assembling procedures, which are referred to in the first paragraph. Please correct us if we were wrong.

Point 5:

The spelt-out difference in the order of component addition for cryo-EM sample preparation was helpful. Based on this order description, RNGTT was added before NELF. BUT doesn't this contradict the statement/conclusion that paused Pol II promotes capping? Meaning, don't you need NELF to first bind Pol II to get it into what the authors assume is a paused state?

Based on our results, we haven't made the statement/conclusion that "paused Pol II promotes capping" in our manuscript. For the glycerol gradient centrifugation in the manuscript, the results showed that the phosphorylation of CTD of Pol II and DSIF facilitated RNGTT recruitment. In the guanylation assay of previous point-by-point response, DSIF promotes the guanylation activity of RNGTT, rather than NELF, suggesting that it is not the paused Pol II promotes capping.

We also set up the extension assay showing that RNGTT did not influence RNA extension. The addition order of RNGTT and NELF didn't alter the NELF-mediated RNA pausing. We have tried the three assembly orders, and decided the method of the paper by Cryo-EM test.

The asterisks represent the last addition and the black dots represent the simultaneous addition.

Points 6 and 7:

In the revised text, authors added statements like "Our structures unravel that capping enzymes RNGTT and CMTR1 can directly bind the paused elongation complex, and shed light on how pre-mRNA 5'-end capping couples with transcription at the pausing stage" (Abstract) and "We show the structural evidence that capping enzymes coexist with pausing factors, which are helpful in understanding the correlation of 5' end capping modifications and Pol II transcription processes" (lines 93-94). Yet, in the response they say that there is no connection between capping activities and NELF states (up/down, paused/unpaused Pol II... although the "up" NELF is yet to be biochemically linked to paused Pol II) or NELF/capping

enzyme association with Pol II.

This brings us back to novelty of the current study: what new is learned? As we see it, what distinguishes this current study from Garg et al. is that here the authors captured NELF in complex with Pol II and capping enzymes (RNGTT). But how exactly does this add to the mechanistic understanding of the link between transcriptional pausing and mRNA capping?

We think the two statements are not contradictory. In previous response to Q7, we stated that there is no connection between capping activities and NELF states, and our assay shows that NELF has no influence on guanylation activity in our assay, irrespective of its states. On the contrary, RNGTT binding to the stalk of Pol II affects NELF conformational change, in which we separated two states through cryo-EM.

In our newly performed extension assay, 5' FAM-labeled RNA is used in extension assay. Without 5' end guanylation of RNA, RNGTT can moderately relieve the repression effect caused by NELF. We speculate that RNGTT can promote pause release in an activity-independent manner, consistent with previous study^[3].

Point 9:

With all due respect, the writing should still be significantly improved, including that in the added text. We strongly recommend seeking assistance of a writing center or such resource to bring writing/storytelling quality up to Nat Commun level.

Thanks for the comments. We have polished the manuscript and hoped that the revised manuscript is acceptable for publication.

We were satisfied with the authors' responses to the rest of our original review. We appreciate the authors performing additional experiments, e.g., guanylation with intermediate RNA lengths, and adding "big picture"/conclusion statements throughout.

Reviewer #3 (Remarks to the Author):

Reviewer #4 (Remarks to the Author):

Within the revised manuscript the authors now provide a much-improved comparison of their PEC-RNGTT-CMTR1 structure to transcribing Pol II-DSIF-RNGTT-CMTR1 and Pol II-DSIF - CMTR1. Also, most other points have been addressed reasonably now. In particular, the discrepancy in optimum RNA length in the capping assay, which was a concern raised by reviewers #1 and #2 is now addressed and discussed based on the improved structural comparison shown in Fig. S6 that now also visualizes the RNA trajectory to the CE. All in all, important aspects emerge in from the manuscript that set this study apart from the Cramer lab's paper published last year ("Down" conformation of NELF, different capping length optimum). It is up to the editor to decide whether this is significant enough to justify publication in Nature communications.

Further points:

Fig. 4c/d is never referenced in the text.

We thank the reviewer for the positive comments. Fig. 4d was referenced in the text. We referenced the Fig. 4c in the revised manuscript.

Line 281-282: "and AlphaFold2 model of GTase-like domain into CMTR1 cryo-EM map (Fig. 4a-4c and Supplementary Fig. 3h)"

- [1] BERNECKY C, PLITZKO J M, CRAMER P. Structure of a transcribing RNA polymerase II–DSIF complex reveals a multidentate DNA–RNA clamp [J]. Nature Structural & Molecular Biology, 2017, 24(10): 809–15.
- [2] VOS S M, FARNUNG L, URLAUB H, et al. Structure of paused transcription complex Pol II–DSIF–NELF [J]. Nature, 2018, 560(7720): 601–6.
- [3] MANDAL S S, CHU C, WADA T, et al. Functional interactions of RNA-capping enzyme with factors that positively and negatively regulate promoter escape by RNA polymerase II [J]. Proceedings of the National Academy of Sciences, 2004, 101(20): 7572–7.

REVIEWERS' COMMENTS

Reviewer #2 (Remarks to the Author):

The authors did their best to address our concerns. We don't have any further comments.

Reviewer #3 (Remarks to the Author):
